# Light scattering and microphysical properties of atmospheric bullet rosette ice crystals

Shawn W. Wagner[1,*], Martin Schnaiter[1,2,**], Guanglang Xu[1,***], Franziska Nehlert[1], and Emma Järvinen[1,**]

[1]Karlsruhe Institute of Technology, Karlsruhe, Germany
[2]schnaiTEC GmbH, Wuppertal, Germany
[*]Now at University of North Dakota, Grand Forks, United States
[**]Now at Institute for Atmospheric and Environmental Research, University of Wuppertal, Wuppertal, Germany
[***]Now at Shenzhen University, Shenzhen, China

**Correspondence:** Shawn W. Wagner (shawn.wagner@und.edu), and/or Emma Järvinen (jaervinen@uni-wuppertal.de)

**Abstract.** Cirrus clouds play a critical role in the Earth's radiative budget. The extent of the radiative impact of cirrus clouds is governed by the physical properties of aspherical ice crystals. One of the most relevant cirrus ice crystal habits is a polycrystalline bullet rosette, where individual bullets radiate from the same nucleation point. Here, the link between the crystal morphology of atmospheric bullet rosettes and their radiative properties in the form of the asymmetry parameter ($g$) is experimentally investigated using correlated high-resolution in situ stereo images of individual rosettes and their corresponding angular scattering functions measured by the airborne Particle Habit Imaging and Polar Scattering (PHIPS) cloud probe. Bullet rosette stereo images are analyzed for their microphysical properties, including maximum dimension, bullet aspect ratio, number of bullets, projected area, bullet hollowness, derived mass, derived effective density, and derived terminal velocity, as well as their optical properties such as $g$ and the optical complexity parameter. Results indicate that much lower $g$ values represent real atmospheric bullet rosette crystals than what is expected by numerical calculations assuming solid or hollow bullets, indicating higher levels of crystal complexity than have been incorporated within previous bullet rosette ray-tracing studies. Measured $g$ values herein have a direct impact on modeling the shortwave reflection of cirrus clouds, resulting in an increase in scaled optical depth by an average 53 % in comparison to previously calculated $g$ values. This study will provide a valuable in situ dataset that can be used as a basis for development of future ice optical models.

## 1 Introduction

Cirrus clouds have been found to cover the Earth's surface at a global average of up to 50 % (Wylie et al., 1994; Lynch, 1996; Sassen et al., 2008; Stubenrauch et al., 2013, 2017). Due to their high frequency of occurrence, as well as being one of the first solid encounters of solar radiation when entering the Earth's atmosphere, cirrus clouds have a major impact on the Earth's radiative budget (Paltridge, 1980; Liou, 1986; Baran, 2012; Yang et al., 2015). The extent of this impact is determined, in part, by the aspherical ice crystal single scattering properties within the cirrus. Therefore, an accurate understanding of the crystal

single scattering properties is necessary to understand the full extent of this impact, which in turn improves climatological and radiative modeling capabilities.

One of the most comprehensive and widely accepted habit diagrams at present, produced by Bailey and Hallet (2009), suggests that the low temperatures and range of supersaturations found within cirrus clouds are ideal conditions for forming the polycrystalline ice habit known as a bullet rosette (Heymsfield and Iaquinta, 2000; Heymsfield et al., 2002). First reported in 1947 by Weikmann (1947), bullet rosettes have since been the subject of numerous studies, facilitated by significant advances in in situ measurement techniques. These include works by Lawson et al. (2006, 2010, 2019), Um and McFarquhar (2007), Um et al. (2015), and many others. The current understanding is that bullet rosettes and other rosette based polycrystals comprise over half of the contents of in situ formed cirrus clouds (Lawson et al., 2019).

Um et al. (2015) conducted an in-depth microphysical analysis of several ice crystals using the Cloud Particle Imager (CPI) and data sets from three campaigns. It was found that bullet rosettes most frequently formed at temperatures of approximately -45 °C and had a mean number of bullets per rosette at $5.5 \pm 1.35$ with an increasing bullet aspect ratio as the number of bullets per rosette increased. The mean maximum dimensions of bullet rosettes ranged from 80 µm at -65 °C to 300 µm at -30 °C, with dimensions generally increasing with increasing temperature. A number of studies have also been conducted to quantify bullet rosette mass and effective density, which is defined as the particle mass divided by the volume of an equivalent diameter sphere (see Heymsfield et al., 2004) (Brown and Francis, 1995; Heymsfield et al., 2002; Baker and Lawson, 2006; Um and McFarquhar, 2011; Erfani and Mitchell, 2016; Fridlind et al., 2016). Fridlind et al. (2016) found bullet rosette masses ranging from 0.001 to 0.1 mg and effective densities ranging from 0.1 to 0.5 g cm$^{-3}$ for maximum dimensions ranging from 200 to 1100 µm. Terminal velocities up to 210 cm s$^{-1}$ were found for a maximum dimension of 1000 µm. This study aims to further build on previous microphysical studies utilizing high resolution cloud particle imagers to increase the accuracy of the bullet rosette microphysical parameterizations.

Particle images taken across the multitude of studies on bullet rosettes show an inherently complex microphysical structure which directly impacts their single scattering properties. Yet, previous studies have been performed to analyze the shortwave optical properties of bullet rosettes using idealized ray-tracing models, such as Iaquinta et al. (1995), Schmitt et al. (2006), and Yang et al. (2008). Iaquinta et al. (1995) found that, assuming a random spatial orientation, the number and configuration of bullets per rosette have little effect on the overall phase function and thus for the asymmetry parameter ($g$). Bullet rosette $g$ values were primarily altered by varying the aspect ratio of the individual bullets, resulting in a range of $g$ from 0.788 to 0.876. Schmitt et al. (2006) extended this work to include bullet rosettes with varying degrees of hollowness. It was determined that "...hollow bullet rosettes have distinctly different scattering properties than do solid bullet rosettes". With compact (aspect ratio = 1) solid bullets, there was found to be a 13 % difference in $g$ versus when the bullets are entirely hollow with $g$ increasing with hollowness. For more elongated (aspect ratio = 0.1) bullets this difference is reduced to 3.5 % with $g$ decreasing with hollowness. Yang et al. (2008) examined both solid and hollow rosettes with updated geometries and using the Improved Geometric Optics Method (IGOM; Yang and liou (1996)). While Yang et al. (2008) reported similar phase functions for solid rosettes as compared to Iaquinta et al. (1995), smoother phase functions for hollow rosettes were seen compared to Schmitt et al. (2006).

In radiative transfer and climatological modeling, $g$ is a crucial component. Thus, an understanding of $g$ associated with real ice crystal habits is critical for a proper simulation of cirrus cloud effects on the Earth's radiative budget. Here we present for the first time in situ single particle polar nephelometer measurements of atmospheric bullet rosette crystals, thus expanding from theoretical calculations to atmospheric observations. In situ atmospheric bullet rosettes were measured during the Cirrus in High Latitudes (CIRRUS-HL) airborne campaign. The observations are compared to numerical ray-tracing results of bullet rosettes to give recommendations to the degree of surface roughness needed to reproduce the observed $g$ values.

## 2 Methodology

### 2.1 Airborne Measurements

CIRRUS-HL was an airborne mission based in Oberpfaffenhofen, Germany, to measure microphysical and radiative properties relating to high-latitude cirrus clouds using the High Altitude and Long Range (HALO) research aircraft. While the focus was intended to be northerly by sampling in subarctic Europe, pandemic restrictions in the base location required the focus to shift towards Central and Southern Europe. Both in situ and liquid origin cirrus (Krämer et al., 2016), as well as aviation related contrail cirrus were targeted. CIRRUS-HL consisted of twenty-four flights (one test flight, one calibration flight, and twenty-two research flights) from June 6th to July 28th 2021, accumulating a total of 140 research flight hours; twenty-one of the twenty-two research flights showed the presence of bullet rosettes. A suite of instruments for cloud microphysical and atmospheric state variables was deployed during CIRRUS-HL (DLR, 2024), including the Particle Habit Imaging and Polar Scattering (PHIPS) cloud probe.

To experimentally investigate the habit-specific angular light scattering properties of atmospheric ice crystals, a single-particle nephelometer is combined with a high-resolution imaging system in PHIPS (Abdelmonem et al., 2016; Schnaiter et al., 2018). This setup allows for the identification and selection of ice crystals with specific habits, such as bullet rosettes, from an ensemble of crystals measured during flight legs in natural cirrus clouds. By isolating the angular scattering functions attributed to a specific habit or microphysical feature, it is possible to derive the representative orientation averaged angular scattering function of this specific habit. In essence, this analysis method generates a cloud composed solely of ice crystals with a specific habit or microphysical feature (e.g., bullet rosettes) based on data from real atmospheric ice crystals.

Upon entering the $\approx 0.3~\mathrm{mm}^3$ detection volume of the PHIPS, a cloud particle scatters light of a continuous wave green laser beam with a wavelength of 532 nm which is registered by the trigger system. Twenty off-axis parabolic mirrors redirect scattered light to corresponding channel optical fibers in the 18° to 170° range, with an 8° resolution. Each mirror has a diameter of 10 mm and is positioned 83 mm from the sample volume, giving a solid angle of 0.011 sr. A multi-anode photomultiplier tube (MAPMT) converts received light into a voltage pulse. The pulse is then digitized to a dynamic range of 2047 counts to provide the measured light scattering intensity for each angular position or the differential scattering cross section per solid angle. A maximum sample rate of 3.5 kHz allows for rapid data acquisition in environments with high cloud particle concentrations.

Simultaneously, two charge-coupled device (CCD) cameras with telescopic assemblies record 1360 by 1024 pixel brightfield images of the cloud particle at a maximum acquisition rate of 10 Hz. The camera assemblies are positioned on either side of the sample volume with a 60° viewing angle of the particle (120° between the cameras). For the magnification settings used in CIRRUS-HL, focused microscopes provide a pixel resolution of $\approx$ 1.61 µm with a maximum particle size range between 1650 - 2200 µm depending on particle orientation. Illumination is provided for 10 ns by an incoherent 690 nm pulsed diode laser which eliminates wave interference and allows for clear, high resolution images. It is important to note that the resolution is limited by the lens system rather than the pixel size, resulting in an optical resolution of approximately 6 $\mu m$ at the magnification used in CIRRUS-HL. For further details on the PHIPS physical design and principles of operation, see Abdelmonem et al. (2016). For information on the PHIPS characterization and initial results, see Schnaiter et al. (2018).

### 2.1.1 Microphysical Analysis

PHIPS stereo-image pairs from CIRRUS-HL are manually reviewed and each imaged ice crystal is classified by their respective habit. Bullet rosettes are then selected for use in this study by assessing their image clarity, and if the crystal has been fully captured by at least one camera. Of the 5668 total bullet rosettes encountered during CIRRUS-HL, 4512 rosettes are accepted for this study. These bullet rosettes were entirely captured in at least one of the two PHIPS camera focal planes allowing for measurement of the maximum dimension with a high confidence. Of the 4512 bullet rosettes accepted, 1292 are found to have individually identifiable and distinguishable bullets, allowing for further analysis of the bullet related microphysical properties to be performed. These 1292 bullet rosettes will be the primary focus of the microphysical results and discussion.

Within the CIRRUS-HL data set, there were 665 bullet rosette aggregates which were fully captured by both cameras. Aggregates were most frequently observed with large rosettes and seemingly the result of collisions between bullets. While aggregates have been excluded for the purposes of this study, they could be the basis for potential future work.

Additionally, of the 5668 bullet rosettes encountered, a criterion was set to select rosettes with angular scattering functions in which no saturation of the first two PHIPS measurement channels occurred. This criterion was chosen since the first two measurement angles are decisive for accurate retrievals of $g$. Limiting the dataset accordingly resulted in a set of 1549 rosettes, only 1.4 % of which had at least one saturated channel between 34° and 170°. Since $g$ retrievals are less sensitive to saturation of side scattering angles, these rosettes will be the focus of the single-scattering properties results and discussion in Sect. 3.2.

Bullet rosettes are further separated into categories based on the presence of cavities or air pockets. These subcategories are: solid, hollow, and inclusions. Examples of each category can be seen in Fig. 1. Solid bullet rosettes are defined as rosettes with bullets having no visible cavities or air pockets; hollow bullet rosettes have bullets with cavities that begin at the outward end of the bullets and move inward forming a convex opening which can span any length of the affected bullet (Schmitt and Heymsfield, 2007; Yang et al., 2008); inclusion rosettes have bullets which contain pockets that do not breach the outward ends of the bullets. Despite careful visible analysis of each available bullet rosette, not all images provide enough detail to determine if a bullet rosette is able to be placed into one of the aforementioned categories. While those rosettes are still included in the overarching group of all rosettes, they are excluded from further categorization. An example of a bullet rosette with an unknown categorization can also be seen in Fig. 1. These categories are specifically important for understanding the single-scattering

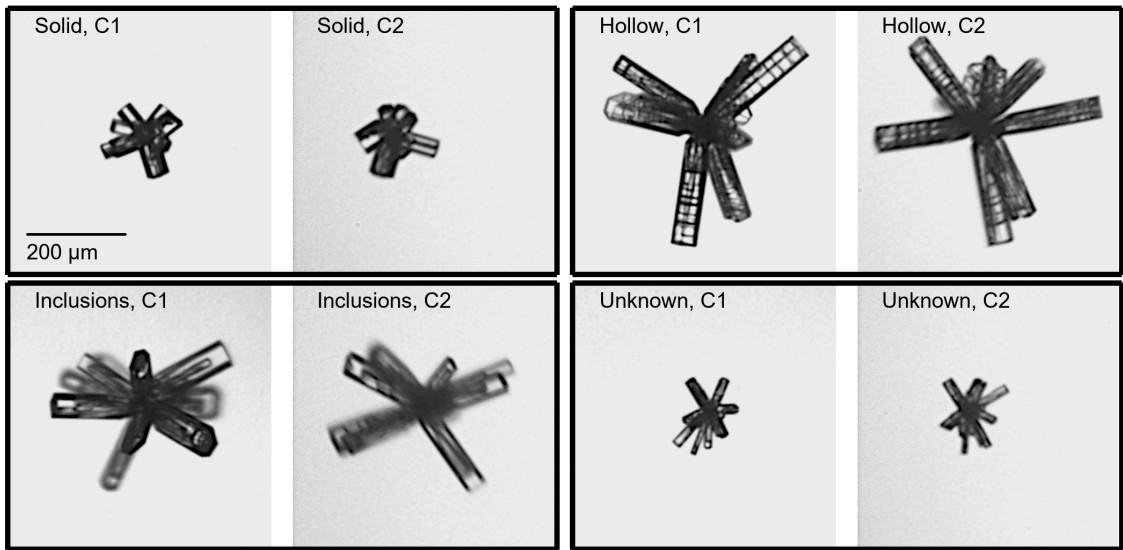

**Figure 1.** Example bullet rosettes imaged by the Particle Habit Imaging and Polar Scattering (PHIPS) probe, separated into categories based on the presence of cavities or air pockets. Camera assembly one (C1) images are on the left, and camera assembly two (C2) images are on the immediate right. There is a 120° separation between C1 and C2.

properties results and discussion. If a rosette contains elements of multiple categories (such as both hollowness and inclusions), that rosette is only included in the overarching group of all rosettes.

Microphysical analysis of the individual bullets is conducted manually using MATLAB GUI based software to display the rosette images. Using the software, bullet number per rosette and the pixel locations of the individual bullet endpoints in terms of both length and width are visually determined and selected. Bullet length ($L_B$) is defined as the number of pixels between the rosette origin and the outward end of the bullet. Bullet width ($W_B$) is the number of pixels between the opposing bullet facets. Individual bullet aspect ratios ($AR_B$) are defined as:

$$AR_B = \frac{W_B}{L_B}. \tag{1}$$

Corrections which account for errors due to $L_B$ and $W_B$ being assessed from bullet rosette projections on 2-D images are addressed in Appendix B. The length of the hollowness for each applicable bullet is visually determined via pixel location using the same method applied to the bullet length and width, where the beginning of hollowness is the edge of the bullet, and the end of the hollowness is some point within the bullet. The hollowness factor is then calculated using the method of Schmitt et al. (2006).

$$H_{FACTOR} = \frac{L_H}{L_B}, \tag{2}$$

where $L_H$ is the length of the hollowness. Each bullet rosette with observable hollowness is represented by a single $H_{FACTOR}$ which is averaged from each hollow bullet.

The maximum dimension ($D_{max}$, defined as the longest possible axis between two darkened pixels) of each bullet rosette is included in the microphysical analysis using IDL image analyzing software that is applied during the primary data processing, and has no corrections applied (Schön et al., 2011). The mass of each rosette is calculated by treating each bullet as a hexagonal column with a hexacone cap, with the $H_{FACTOR}$ accounted for when applicable. Bullet rosette mass is represented as:

$$m = \rho_{ice} \times \overline{V_B} \times N_B, \tag{3}$$

where $\rho_{ice}$ is the density of solid ice and is assumed to be 0.917 g cm$^{-3}$, $\overline{V_B}$ is the average bullet volume for that rosette, and $N_B$ is the number of bullets. Detailed equations pertaining to $\overline{V_B}$ can be found within Appendix B. Using the calculated bullet rosette mass, the effective density of each rosette is calculated using:

$$\rho_e = \frac{6m}{\pi D_{max}^3}. \tag{4}$$

Using methods outlined in Mitchell (1996), calculated bullet rosette mass with the measured bullet rosette projected area ($A$) can also be used to calculate the terminal velocity of each rosette:

$$V_t = \sqrt{\frac{2mg_c}{\rho_a A C_D}}, \tag{5}$$

where $g_c$ is the gravitational constant (9.81 m s$^{-2}$), $\rho_a$ is the density of air and $C_D$ is the drag coefficient. Detailed equations pertaining to $\rho_a$ and $C_D$ can be found within Appendix C.

### 2.1.2 Single-Scattering Property Retrievals

As with any polar nephelometer, calculating $g$ from PHIPS measurements requires a retrieval algorithm to account for the limited angular range of the instrument. The methods applied in this study have been thoroughly described in Xu et al. (2022a). In short, the method is based on the assumption that the asymmetry parameter in geometrical optics regime can be divided into a diffraction part and geometrical optics part (Macke et al., 1996). The diffraction phase function contributes mainly outside the PHIPS measurement range in the forward direction so that the diffraction $g$ can be estimated based on particle size. The geometrical optics $g$ is derived from the PHIPS measurements by fitting the data with Legendre polynomials. When we later report $g$, the values represent the combined $g$, incorporating both the diffraction and geometrical optics components.

In the process of characterizing the accuracy of the $g$ retrievals, Xu et al. (2022a) introduced an optical complexity parameter ($C_p$): a description of the isotropic degree of an angular scattering function which strongly correlates to $g$ (Xu et al., 2023). This can be calculated using (Xu et al., 2022b):

$$C_p = (\sum_{l=0}^{\infty} |\hat{c}_{GO,l}|)^{-1}, \tag{6}$$

where $\hat{c}_{GO,l}$ is the expansion coefficients of the geometric optics phase function using a series of Legendre polynomials. These expansion coefficients are used in the $g$ retrieval as well.

While $g$ can range from -1 to 1, $C_p$ only ranges from 0 to 1; 0 indicates a delta function and 1 indicates perfectly isotropic scattering. Thus, the angular scattering function becomes more featureless with increasing values of $C_p$. Xu et al. (2022b)

showed that $C_\mathrm{p}$ correlates with the complexity parameters used in ray-tracing models, and therefore we refer to it as an optical complexity parameter. A summarized case study of $C_\mathrm{p}$ as applied to rimed particles measured by the PHIPS can be found in Xu et al. (2022b). It should be noted that $C_\mathrm{p}$ is derived from the PHIPS measured angular scattering function, not from the stereo-images. As will be shown in Sect. 3.2, a $C_\mathrm{p}$ - $g$ relation is useful when comparing the PHIPS measurements to the results of optical models.

## 2.2 Numerical Simulations

Numerical ray-tracing simulations are performed to compare the PHIPS bullet rosette $g$ measurements from CIRRUS-HL to those of modeled rosettes, using solid rosette, hollow rosette and solid column geometries with the geometrical optics method of Macke et al. (1996). Crystal complexity is incorporated using the uniform tilted angle (UTA) method of Macke et al. (1996) to simulate crystal distortion, with distortion parameters ($\delta$) including 0.0 (no distortion), 0.3, 0.6 and 0.9 (high distortion). It should be kept in mind that a $\delta$ value of 0.9 corresponds to an extreme shape distortion, which likely is an unphysical shape for natural ice crystals. Each crystal distortion was applied to ray-tracing simulations for the three crystal habits modeled for five area equivalent diameters ($D$) corresponding to the PHIPS size bins: 60 μm, 80 μm, 125 μm, 175 μm, and 225 μm. Additionally, a tilt angle distribution following the frequently utilized Gaussian tilted angle distribution (GTA) (e.g. Liou et al., 1998; Yang and Liou, 1998; Yang et al., 2013; Liu et al., 2013) method is also applied for solid columns. In the case of the GTA approach, the complexity parameter ($\sigma$) is adjusted between 0.1 and 0.9 in increments of 0.1 for each size group. Thus, the total number of ray-tracing simulations is 120. Approximations such as the UTA and GTA methods have previously been shown to closely follow more detailed physical representations of surface roughness and are therefore suitable for use herein. (Liu et al., 2014).

The length and width of the bullets and columns are varied to maintain $AR_B$ of 0.2 while adjusting $D$. Both the solid and hollow bullet rosette simulations are based on a single bullet configuration with seven bullets. Bullets of the hollow rosette each have an $H_{FACTOR}$ of 0.9. An additional form of crystal complexity in numerical ray-tracing can be achieved by simulating the optical effects of internal inclusions. This is done by artificially generating a scattering event after a certain distance, or mean free path (MFP), within the crystal is traveled (Macke et al., 1996). Though not considered for bullet rosettes, simulations of varying MFP were performed for columns. The simulated light wavelength is 532 nm to match that of the PHIPS. A single scattering albedo of 1, ray density of 0.03, 13 internal reflections, and 8 ray recursions with 10 000 crystal orientations are used with all values chosen to minimize the model runtime while maintaining realistic scattering conditions. Randomized orientations provide statistically robust results given that the preferential orientation is not known, and follows the methods of previous studies (Iaquinta et al., 1995; Schmitt et al., 2006; Yang et al., 2008; Um and McFarquhar, 2011; Fridlind et al., 2016). Plots examining differences with varying numbers of orientations for the full range of analyzed sizes, $\delta$, and $\sigma$ are included in a supplement. The standard ice refractive index of 1.31 is also applied (Warren, 1984).

## 3 Results and Discussion

### 3.1 Microphysical Properties

Figure 2 gives an overview of the CIRRUS-HL cloud habit fraction as a function of temperature at the time of the measurement. For the CIRRUS-HL campaign as a whole, bullet rosettes comprise an average 15 % of all cloud particles at temperatures less than -45 °C. As the temperature increases the high frequency of bullet rosettes steadily decreases, and is partly replaced with side planes (polycrystals with plates extending outward from a central point) and mixed rosettes (rosettes containing both columns and plates). This is due to warmer environments (-10 °C > $T$ > -40 °C) being conducive to plate-like growth regimes as opposed to the columnar regime dominate colder temperatures ($T$ < -40 °C) (Bailey and Hallet, 2009). Temperatures from -40 to -50 °C consisted of 11 % bullet rosettes, which agrees well with the Lawson et al. (2019) findings on the frequency of bullet rosettes and bullet rosette polycrystals in clouds of the same temperature. However, the results of this study indicate temperatures of -50 to -60 °C containing 19 % bullet rosettes, while Lawson et al. (2019) found only 10 % in that temperature range. As Lawson et al. (2019) used data from the Airborne Tropical TRopopause EXperiment (ATTREX) which often sampled cirrus from convective outflow, this discrepancy may be the result of the Lawson et al. (2019) crystals having been generated by convective processes rather than in situ origin cirrus. In our study both in situ and convective (liquid) origin cirrus were measured. While the formation of bullet rosettes is most common at temperatures less than -40 °C (Bailey and Hallet, 2009), they were found at temperatures as low as -25 °C. It should be noted that during all but one flight, bullet rosettes occurred at mean temperatures lower than -40 °C and most frequently at temperatures between -45 °C and -50 °C. This strongly agrees with the findings of Um et al. (2015). The relative humidity with respect to ice at the time of sampling varied between 85 % and 145 % (not shown), and does not indicate a strong relation to the presence of bullet rosettes as temperature does.

Stereo examples of the bullet rosettes chosen for analysis can be seen in Fig. 3 with their associated $C_\mathrm{p}$ value (further discussed in the following subsection). The analyzed rosettes have primarily asymmetrical bullet configurations, and frequently there are bullets of varying lengths and $AR_B$ on a single rosette. These variations can also be seen numerically in Fig. 4, which shows the bullet rosette maximum dimension (top left) for all 4512 successfully imaged bullet rosettes with black dots indicating median values for all 1292 rosettes which were accepted for a further microphysical analysis, the bullet $AR_B$ and number of bullets per rosette for the 1292 microphysically analyzed bullet rosettes, and $H_{FACTOR}$ for the bullet rosettes with observable hollowness in the 1292 rosette subset. While the maximum dimensions and number of bullets were taken as the largest values between the two PHIPS cameras because the larger value will always be the most representative, $AR_B$ and $H_{FACTOR}$ are the mean values of the bullets per rosette to act as a representative value. As in Fig. 4, the bullet rosettes were examined in relation to both relative humidity with respect to ice as well as condensable water vapor (not shown). However, no further trends than those about to be outlined with relation to temperature were found.

Bullet rosette maximum dimensions for 4512 accepted rosettes range from 40 µm at a minimum to 1000 µm at a maximum, with the median maximum dimensions spanning from 200 µm to 420 µm. While fairly consistent at temperatures below -35 °C, the median maximum dimension abruptly increases at -35 °C and warmer. The range of medians for only the rosettes accepted for further analysis (indicated with black dots) is higher at 300 µm to 525 µm. This can be explained by a bias owing to larger

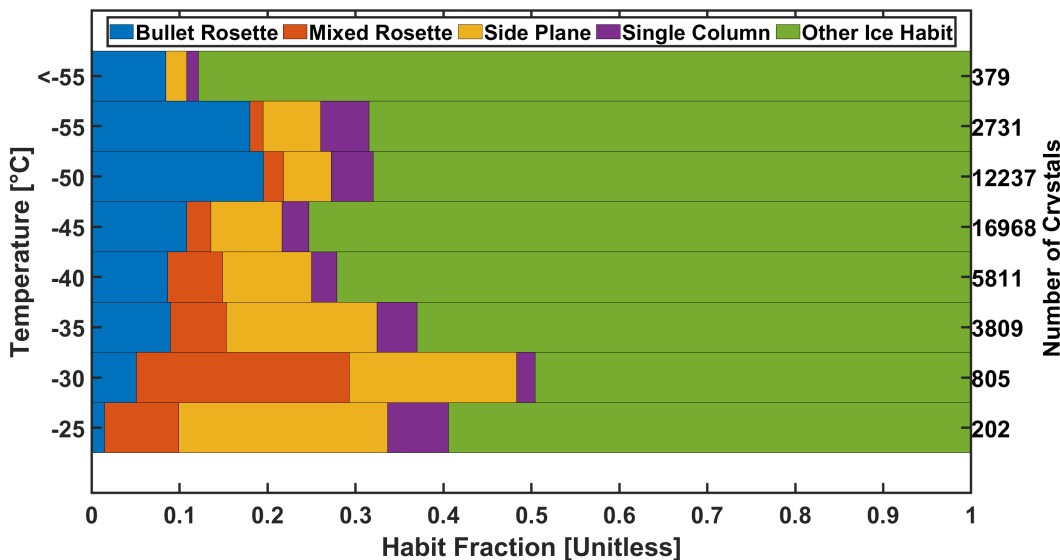

**Figure 2.** Habit fractions of various ice crystals that were both imaged and had their light scattering properties measured by the Particle Habit Imaging and Polar Scattering (PHIPS) probe during the Cirrus in High Latitudes (CIRRUS-HL) airborne campaign. The sampled ice crystals are grouped by temperatures from less than -50 °C to greater than -35 °C in 2.5 °C increments. Numbers on the right y-axis indicate the total number of sampled particles per temperature bin.

rosettes being more likely to have bullets which are confidently analyzable. When considering all acceptable bullet rosettes, the median maximum dimension is equal to or less than 420 μm at all analyzed temperatures, which agrees with Um et al. (2015). $AR_B$ of individual bullets extend from a minimum of 0.1 to a maximum of 0.45, and the median $AR_B$ remains fairly consistent at approximately 0.25 across most observed temperatures. A source of uncertainty for the visually based analysis is the potential inaccuracy in manually selecting the pixels for the outer edge and inner point of the individual bullets. Naturally, the objective is to be as close as possible to the outer edge of the bullet as well as the central origin point within the rosette. While identifying the outer edge is rather simple, Figs. 1 and 3 show how the point of origin is rarely well defined and often leads to an estimation as to the pixel location. However, the estimation that is made is assumed to be fairly reliable as examining the position of all bullets should lead to a more or less obvious conclusion as to the location of the rosette center. To quantify this uncertainty, ten bullet rosettes are chosen at random and a bullet length analysis is performed five times on each rosette. The relative standard deviation between the tests is found to be 2.7 %.

While there is a high variation in the number of bullets that are observed from three to as high over twelve, the median is consistently between seven and eight across all temperatures of the analysis. This is only slightly higher than the four to seven mean number of bullets reported by Um et al. (2015). This discrepancy can be explained by Um et al. (2015) analyzing images from only one viewing angle, causing some bullets in the background to be obscured by those in the foreground and thus under-representing of the number of bullets. However, even with stereo imaging the complex three-dimensional nature of

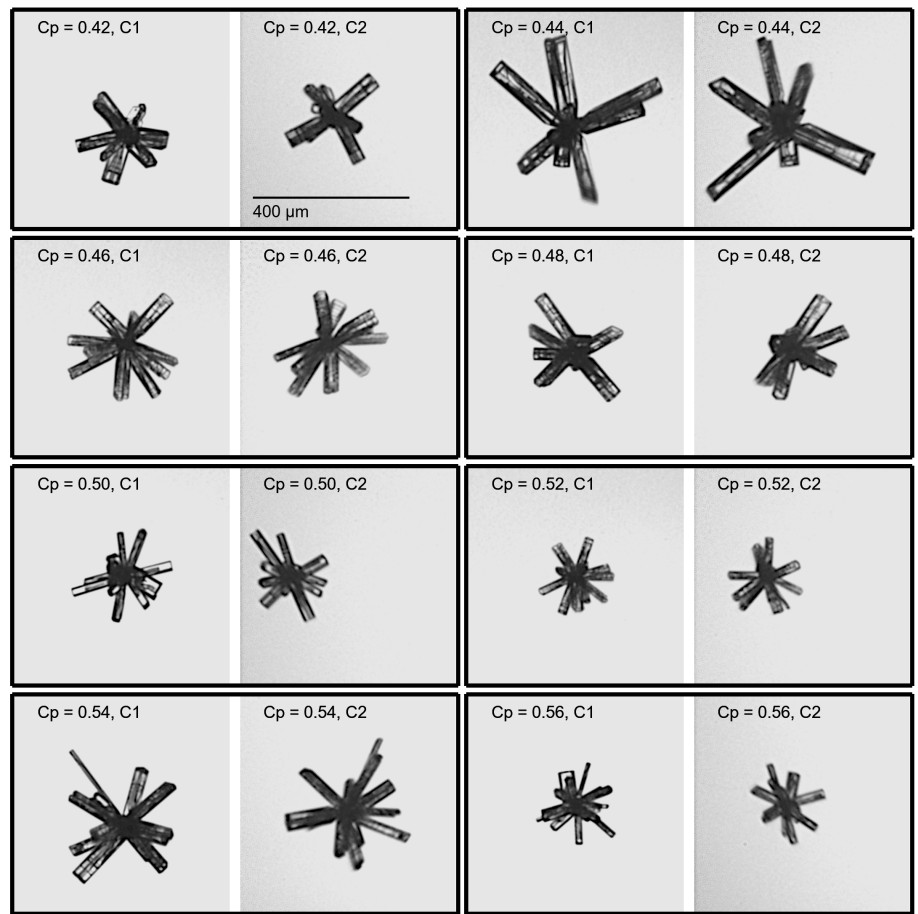

**Figure 3.** Bullet rosettes sampled during the Cirrus in High Latitudes (CIRRUS-HL) airborne campaign by the Particle Habit Imaging and Polar Scattering (PHIPS) probe, with camera assembly one (C1) images on the left, and camera assembly two (C2) images on the immediate right. There is a 120° separation between C1 and C2. The images are sorted in order of increasing complexity parameter ($C_{\mathrm{p}}$).

bullet rosettes poses complications in accurately identifying the number of bullets present on each rosette. While as a rule all bullet rosettes used for the bullet microphysical analysis have been filtered to only include rosettes where the number of bullets could confidently be stated, it is still possible for bullets in the background to be hidden by those imaged in the foreground. This is true despite the PHIPS stereo images representing viewing angles separated by 120°. The authors find this to be particularly true for rosettes with more than six bullets; beyond twelve bullets an accurate identification of the number is nearly impossible. Thus, the median value of seven to eight bullets per rosettes found here can be under-representative to some degree.

Of the 1292 bullet rosettes chosen for a manual bullet analysis, 932 contained some degree of hollowness. Of those 932, 329 (35 %) have at least one bullet which is not only hollow but has the full extent of the hollowness observable within its associated image. While the $H_{FACTOR}$ reaches as low as 0.2, most bullets with hollowness observed have an $H_{FACTOR}$ between 0.6 and 1.0, indicating extensive hollowness. In comparison, Schmitt and Heymsfield (2007) reported finding rosette

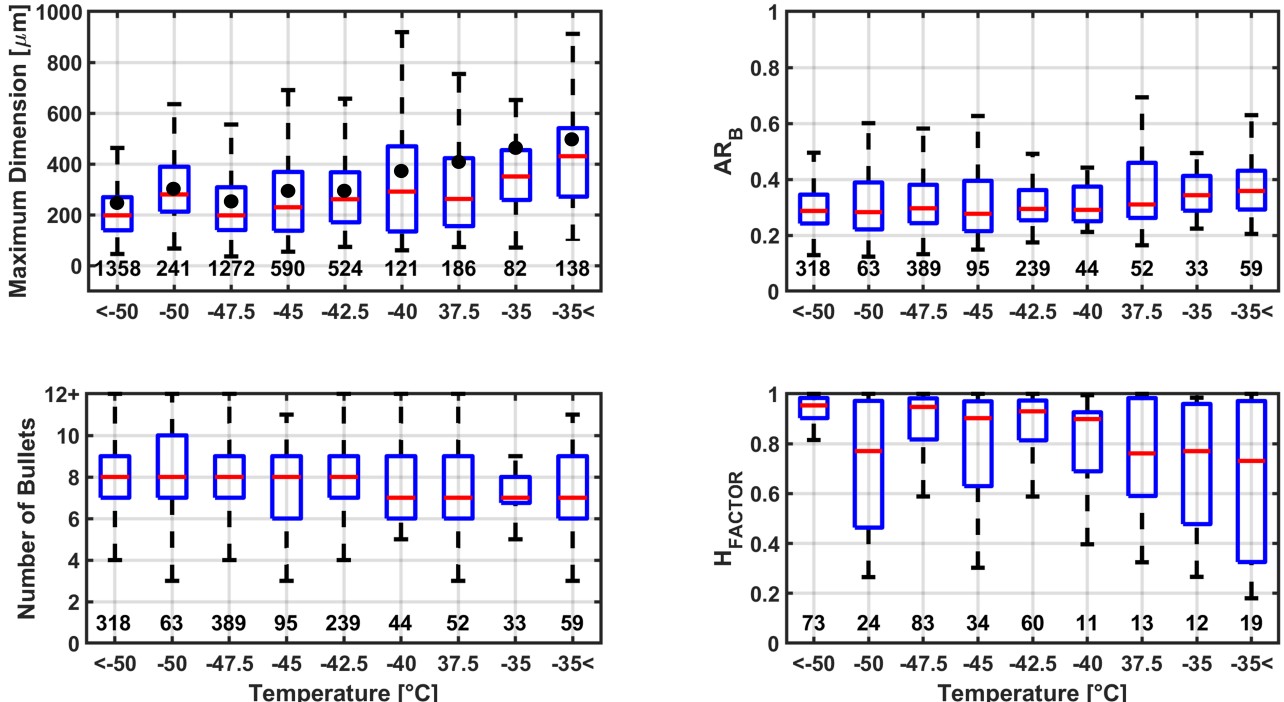

**Figure 4.** The bullet rosette maximum dimension (top left) for all 4512 successfully imaged bullet rosettes. Black dots indicate median values for all 1292 rosettes which were accepted for a further microphysical analysis. The bullet $AR_B$ (top right) and number of bullets per rosette (bottom left) for the 1292 bullet rosettes accepted for a microphysical analysis. $H_{FACTOR}$ of individual bullets (bottom right) for rosettes with observable hollowness taken from the 1292 rosette subset ($H_{FACTOR}$ = 1 is completely hollow). For each plot bullet rosettes are grouped by temperature.

The red horizontal lines within the boxes indicate the median value for that temperature. The values above the x-axis indicate the total number of bullet rosettes for that group.

shaped crystals with hollow bullets to have hollow components extending on average $88 \pm 10$ % the length of the bullet. Just as was mentioned with the calculation of the $AR_B$, there is some uncertainty with the determination of the location of the pixels

relating to $H_{FACTOR}$. As is done to calculate the uncertainty in the bullet length, a hollowness analysis is performed five times on ten randomly selected hollow rosettes. The relative standard deviation between the tests is found to be 4.8 %. While there is a trend toward a decrease in hollowness with increasing temperature, which disagrees with previous literature such as Bailey and Hallet (2009), the lack of statistical robustness makes this result inconclusive.

Figure 5A and Fig. 5B show mean bullet length $L_B$ and width $W_B$ with respect to rosette maximum dimension $D_{max}$ in

265   µm, respectively. Data point colors correspond to $H_{FACTOR}$, with grey squares indicating rosettes for which the degree of hollowness is unmeasurable. Both $L_B$ and $W_B$ have a strong positive correlation ($R^2 = 0.94$ and 0.66, respectively) with $D_{max}$, with $L_B$ often a factor of three to five greater than $W_B$ (Fig. 5C). These findings are consistent with Fridlind et al. (2016).

Figure 5D shows rosette projected area ($A$) with respect to $D_{max}$ with the results of Fridlind et al. (2016) and the Bucky Ball method of Um and McFarquhar (2011) included. The rosette projected area is taken as the maximum between the two PHIPS viewing angles. There is strong agreement between this study and Fridlind et al. (2016) until $D_{max} = 800$ μm when Fridlind et al. (2016) becomes slightly higher, in contrast to the Bucky Ball method which begins to deviate significantly at $D_{max} = 600$ μm.

Despite the similar bullet sizes between this work and Fridlind et al. (2016), there is notable difference relating $m$ to $D_{max}$ (Fig. 6A). While the calculated mass in this study and that of Fridlind et al. (2016) has nearly identical trends with increasing size, the mass of Fridlind et al. (2016) is consistently lower by as much as 35 %. This can be explained by assumptions made on the number of bullets per rosette. In this study each $m$ is calculated using each individual rosette's determined number of bullets, where as Fridlind et al. (2016) assumes each rosette to have six bullets. As shown in Fig. 4, seven to nine bullets are often found on a single rosette. As Fridlind et al. (2016) also does not account for $H_{FACTOR}$, it is apparent that the number of bullets is a stronger deciding factor in $m$ although both are necessary to accurately calculate $m$. The same effect can be seen in relating $m$ to $A$ (Fig. 6B). The parameterizations for relating both $D_{max}$ and $A$ can be found within their respective panel legends.

Both of these effects on $m$ become increasingly significant when applied to the rosette effective density $\rho_e$ (Fig. 7). While rosettes with $D_{max} = 1000$ μm show results higher than those of Fridlind et al. (2016) by approximately 50 %, this reduces to only approximately 10 % higher as $D_{max}$ approaches 100 μm. For $V_t$ the difference in mass has an even stronger effect (Fig. 8). Calculated $V_t$ ranges from 60 to 310 $\mathrm{cm\,s^{-1}}$ for $D_{max}$ between 100 and 900 μm; approximately 50 % higher than the magnitude of values reported by Fridlind et al. (2016). The difference in $V_t$ is unsurprising as $V_t$ is directly dependent on both mass and the projected area. While the projected areas in both this study and Fridlind et al. (2016) are similar, the differences in mass are enough to explain the considerable discrepancy in $V_t$. While beyond the scope of this analysis, the effect of such a difference in $V_t$ would be interesting to explore in modeling applications.

## 3.2 Single-Scattering Properties

Figure 9 shows the PHIPS measured angular scattering functions as differential scattering cross sections, with their associated $g$ and $C_\mathrm{p}$ values and the respective uncertainties for all bullet rosettes, solid rosettes, hollow rosettes, and inclusion rosettes. Each angular scattering function is an average of individual and presumably randomly oriented particle measurements within the categories. Solid circles indicate the PHIPS measurements from the 18° to 170° angles with 8° resolution. The solid black line and dashed colored lines show the retrieved angular scattering functions (without diffraction) from the 0° to 180° angles with 0.06° resolution. The retrieved angular scattering functions are the results of the Legendre polynomial fit applied to the PHIPS measured angular scattering functions.

Hollow and inclusion rosettes show very similar angular scattering functions, with the magnitude of the inclusion rosette angular scattering function increasing only slightly from 60° to 180° relative to the angular scattering function of the hollow rosettes; however, the magnitude of the solid bullet rosettes in the 60° to 80° range has a peak which is not shared with either hollow or inclusion rosettes. This is followed by a slight shift toward the backward direction in the 120° to 140° dip and the 140°

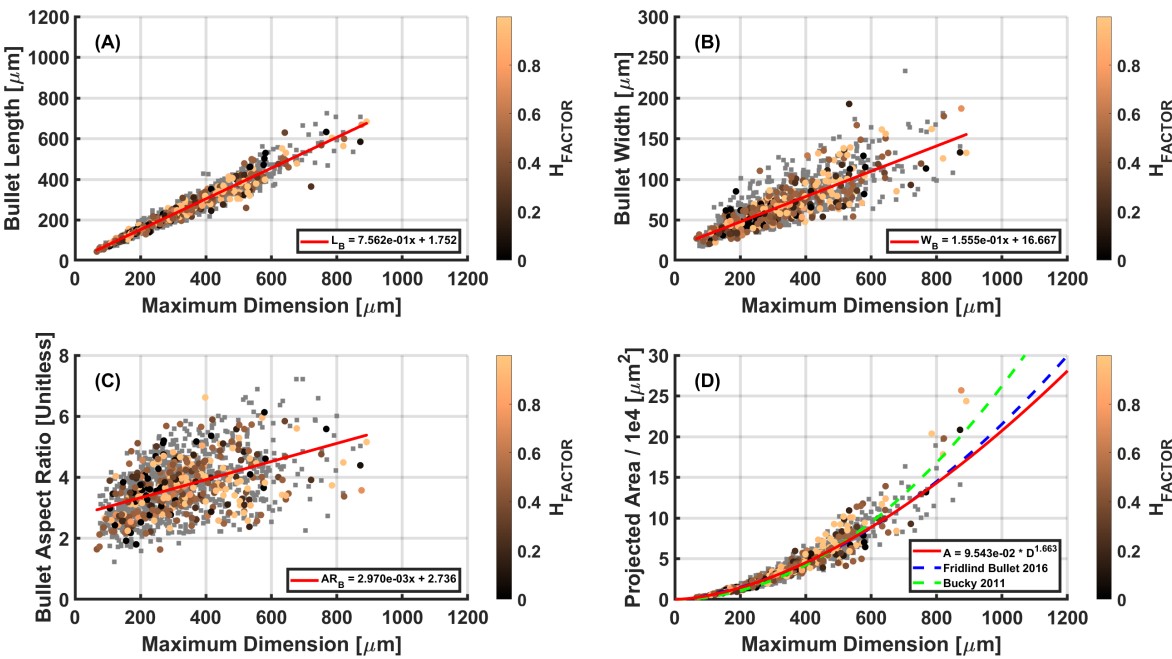

**Figure 5.** Bullet rosette and mean bullet microphysical properties as related to bullet rosette maximum dimension, color coded by the corresponding $H_{FACTOR}$. Rosettes for which the degree of hollowness is unmeasurable are indicated with a grey square.

to 170° peak of the solid rosettes. It should be noted that the number of solid rosettes is significantly lower (33) than rosettes with degrees of hollowness (572) or air inclusions (144), and thus the mean differential cross section is statistically less robust, and an assumption of orientation averaged population is likely not valid as specular reflections of individual particles could have a large affect on the result (see the examples of angular scattering functions measured for individual bullet rosettes in Fig. 9). It should also be noted that the Legendre polynomial fit constitutes a harmonic analysis, meaning that local systematic variations in the measured data are represented by smooth, oscillatory patterns.

PHIPS measurements of each rosette category show a fairly smooth behavior from 0° to 60°. This is in strong contrast to previous modeled results which show rapid drops from the forward most angles followed by several peaks primarily corresponding to the 22° and 46° halos (Iaquinta et al., 1995; Schmitt et al., 2006; Yang et al., 2008). For instance, Schmitt et al. (2006) present theoretical ray-tracing phase functions for both solid and hollow bullet rosettes that assume six bullets, a semi-randomized orientation, $AR_B$ of 0.25, pristine surfaces, and a $H_{FACTOR}$ of 1.0 for the hollow bullet rosette. While the results of Schmitt et al. (2006) show several peaks in angles less than 50° for both the simulated solid and hollow rosettes, indications of these features are entirely missing in the angular scattering functions obtained by the PHIPS. Even though the 22° and 46° halo scattering directions lie precisely between the PHIPS measurement angle pairs 18°/26° and 42°/50°, the absence of any indication of these features in the PHIPS averaged data strongly suggests that they are truly absent in the angular scattering

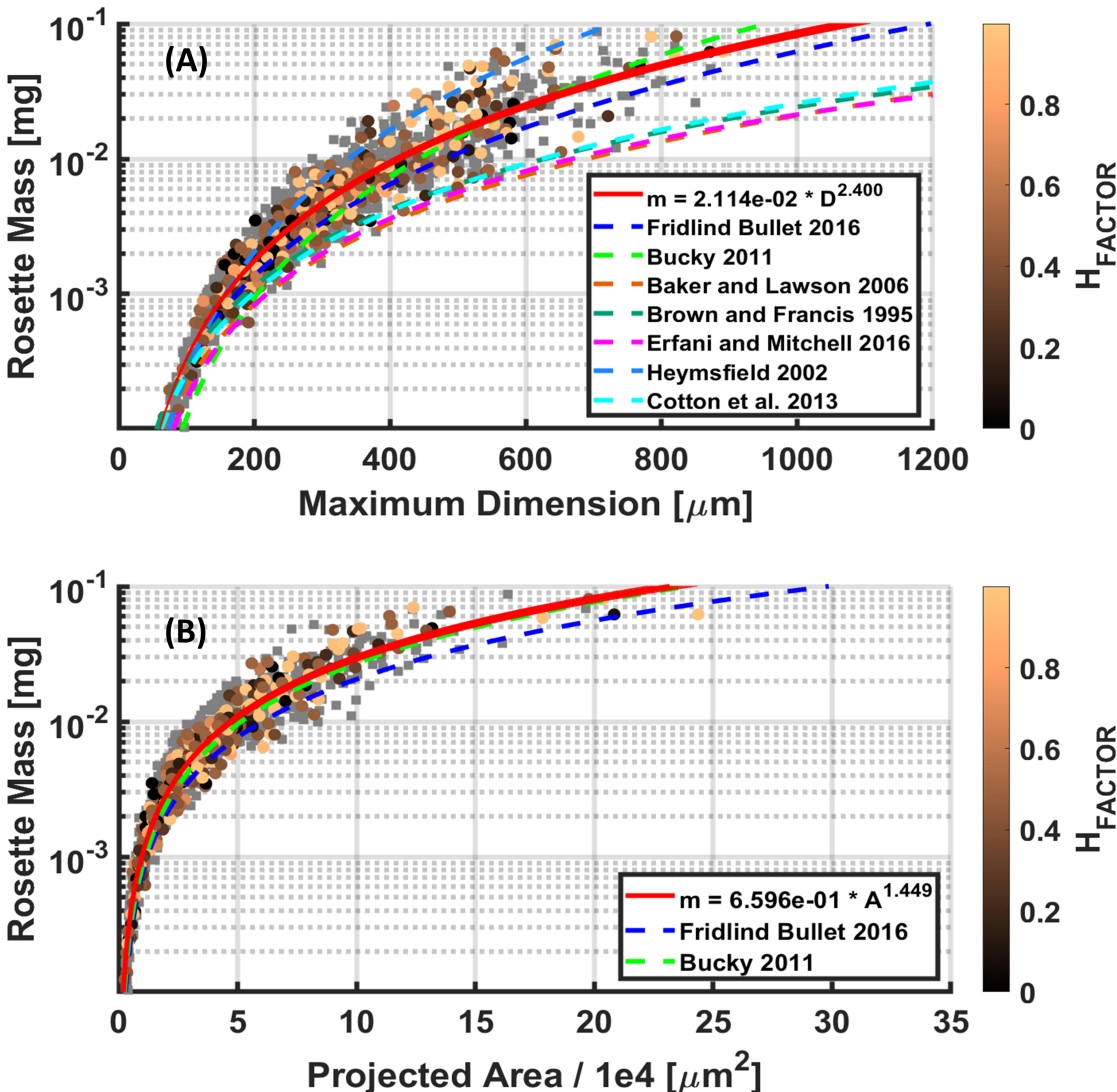

**Figure 6.** Bullet rosette maximum dimension and projected area (*A*) as related to bullet rosette mass, color coded by the corresponding $H_{FACTOR}$. Rosettes for which the degree of hollowness is unmeasurable are indicated with a grey square. Note that a correction to the Brown and Francis (1995) power law been applied to account for a conversion to maximum dimension according to Hogan et al. (2012).

function of real atmospheric bullet rosette ice crystals. This discrepancy can likely be attributed to the theoretical simulations assuming idealized, pristine surfaces while neglecting the natural complexity of ice crystals, a hypothesis that will be further discussed in the following paragraphs.

In the sideward and backward direction, the results of Iaquinta et al. (1995), Schmitt et al. (2006), and Yang et al. (2008) are generally similar. However, Yang et al. (2008) found a steeper decrease and a lower flattening of the scattering phase function

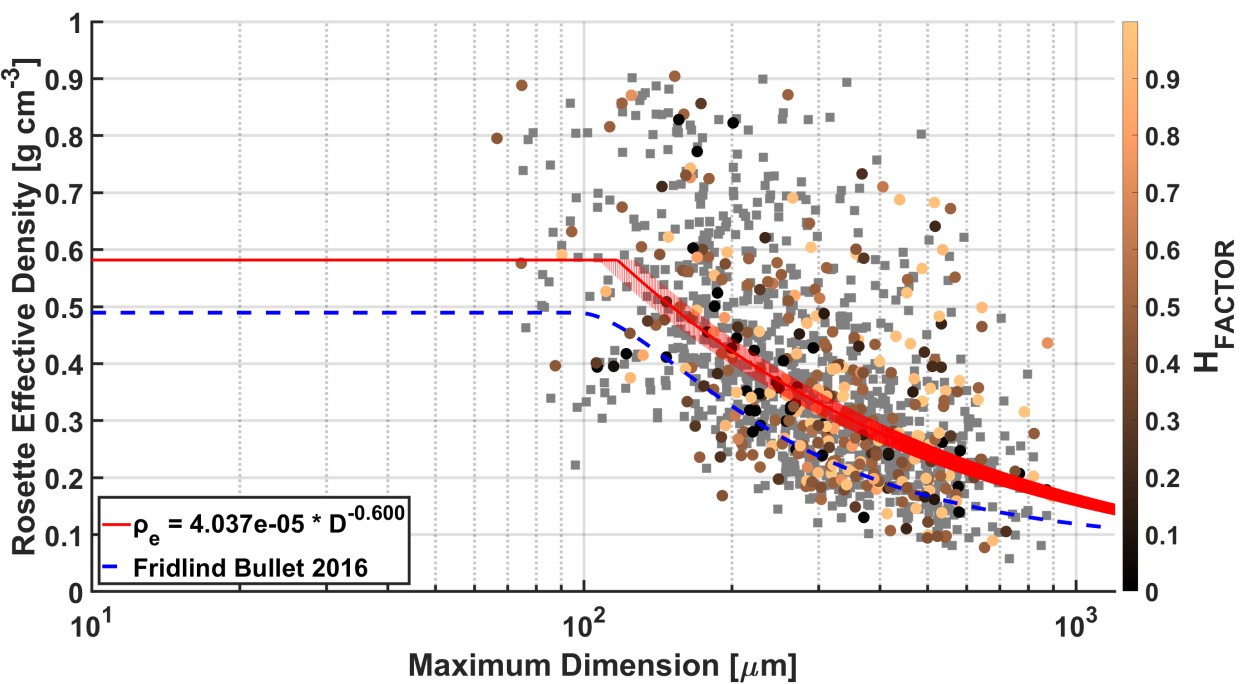

**Figure 7.** Bullet rosette effective density as related to the rosette maximum dimension, color coded by the corresponding $H_{FACTOR}$. Rosettes for which the degree of hollowness is unmeasurable are indicated with a grey square. To account for the natural limit of ice density, values of the effective density parameterization for $D_{max} = < 150$ μm are limited to the calculated mean effective density of the smallest observed rosettes.

toward backscattering directions of hollow bullet rosettes with strong hollowness compared to Schmitt et al. (2006). This behavior aligns more closely with the PHIPS-derived functions plotted in Fig. 9. Theoretical results show a general flattening of the angular scattering functions of both solid and hollow rosettes in the sideward directions. In all categories of the PHIPS measured bullet rosettes there is a continued decrease in the angular scattering function until approximately 130° when the trend reverses and a local maximum is reached for all rosettes, hollow rosettes, and inclusion rosettes at 145°. For solid rosettes these features are slightly shifted to 135° and at 155°, respectively. While this local maximum, sometimes referred to as an "ice bow", is readily apparent in Iaquinta et al. (1995) and Yang et al. (2008), it is less apparent in the results of Schmitt et al. (2006) especially for deeply hollow bullet rosettes.

PHIPS retrieved $g$ values between each category of bullet rosette show an absolute maximum difference of 0.029 (a factor of five larger than the uncertainty) between the lowest values ($g = 0.692$ for inclusion rosettes) and the highest values ($g = 0.721$ for hollow rosettes). When accounting for the uncertainty of 0.006 (more information in Appendix A), this results in a maximum percent difference of 5.8 % between categories. As $C_p$ tends to negatively correlate with positive values of $g$, the lowest value is 0.498 (hollow rosettes) and the highest value is 0.562 (inclusion rosettes). At an absolute maximum difference

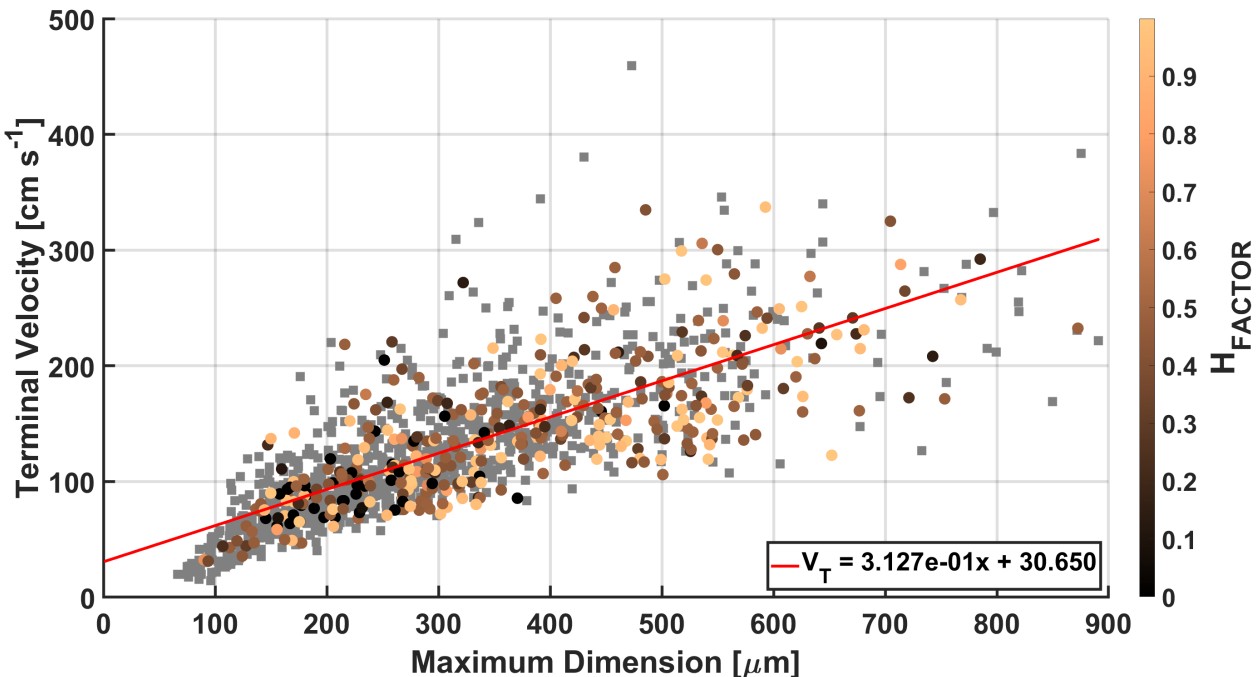

**Figure 8.** Bullet rosette terminal velocities as related to the rosette maximum dimension, color coded by the corresponding $H_{FACTOR}$. Rosettes for which the degree of hollowness is unmeasurable are indicated with a grey square.

of 0.064, there is a maximum percent difference of 16.9 % between categories. While it is evident that the type (solid, hollow, or inclusion) of rosette has little impact on $g$ or $C_p$, other physical properties require consideration. The next critical point of analysis is to relate $g$ and $C_p$ to bullet rosette size.

The symbols in Figure 10 show $g$ versus $C_p$ for all bullet rosettes according to their respective area equivalent diameter ($D$) as measured by the PHIPS imager, grouped by five bin mean sizes: 60, 80, 125, 175 and 225 µm. The asymmetry parameter

$g$ and the complexity parameter $C_p$ are deduced from the angular scattering functions according to the Legendre polynomial analysis given in Sect. 2.1.2. Each $D$ is equal to the diameter of a sphere having the same projected area as the corresponding measured rosettes. As done with the maximum dimensions previously, the largest $D$ between the two PHIPS cameras for each rosette is applied. The mean angular scattering function for the bullet rosettes is calculated for each size group, and that mean angular scattering function is used to retrieve one $g$ and one $C_p$ value for the group. It is assumed that the populations

within these groupings are orientation averaged. Excluding the largest 225 µm bin, Fig. 10 indicates a strong linear negative correlation between $g$ and $D$. Values for $g$ range from 0.751 at $D = 60$ µm to 0.700 at $D = 175$ µm. As Iaquinta et al. (1995) reported theoretical $g$ values ranging from 0.788 to 0.876 for solid rosettes, Schmitt et al. (2006) reported a range from approximately 0.800 to 0.845 for hollow rosettes, and Yang et al. (2008) minimum $g$ values between 0.75 and 0.83 for solid and hollow large rosettes, respectively, there is an approximately 11.5 % mean difference between the now measured $g$ and

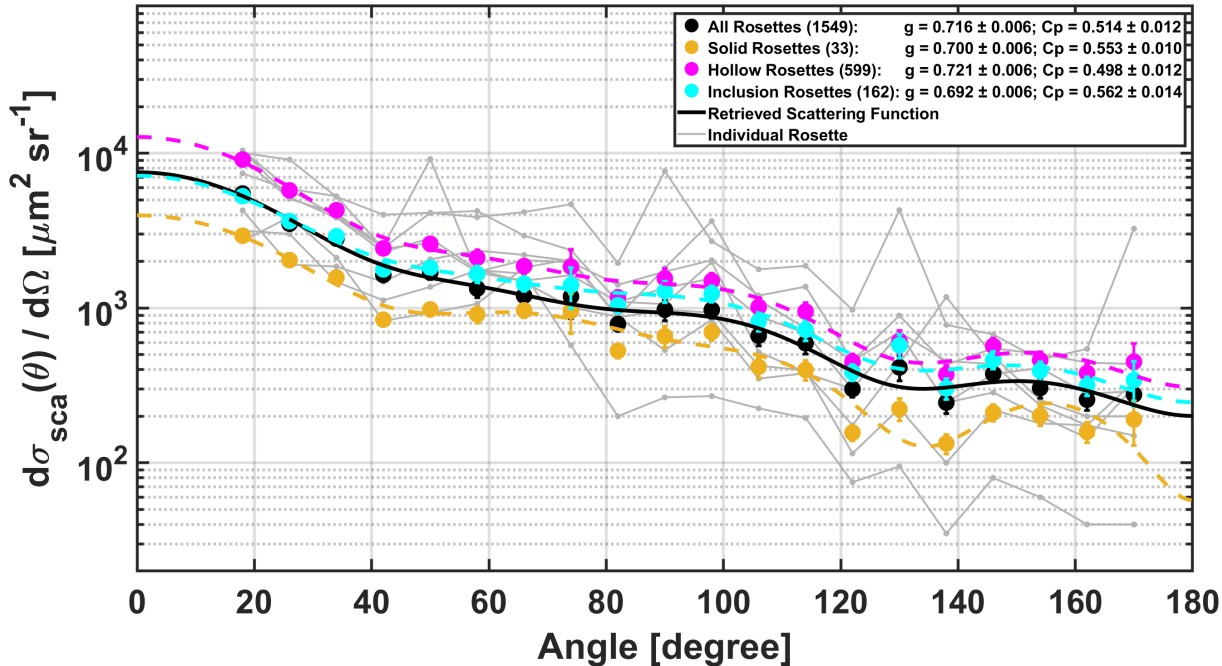

**Figure 9.** The mean bullet rosette angular scattering functions as differential scattering cross sections with their associated asymmetry ($g$) and complexity ($C_p$) parameters and the respective uncertainty separated by bullet rosette category. The values in parentheses indicate the number of rosettes per category. The solid black line and the dashed colored lines show the retrieved angular scattering function used to calculate $g$ and $C_p$. The vertical lines indicate measurement uncertainty. Grey solid lines depict angular scattering functions of the eight individual rosettes shown in Fig. 3

previous theoretical calculations. This difference can be explored by introducing previously omitted surface roughness into new ray-tracing simulations.

To investigate to what extent ice crystal surface roughness can explain the measured low $g$ values, numerical ray tracing simulations with bullet rosette shapes and varying surface roughness metrics were performed. The curves shown in Fig. 10 indicate the range from minimum to maximum theoretical $g$ by varying simulated roughness parameters (distortion parameter $\delta$ in UTA and complexity parameter $\sigma$ in GTA, see Sect. 2.2) of the ray-tracing simulations for solid and hollow rosettes with the same $D$ bins as those of the PHIPS ($D$ variation is shown as green and blue shaded areas in Fig. 10). The assumed rosette geometry is discussed in Sect. 2.2, with the hollow bullet rosettes having an $H_{FACTOR}$ of 0.9 applied.

With little to no simulated roughness applied (when $C_p < 0.45$), the modeled values of $g$ are similar to those of Iaquinta et al. (1995), Schmitt et al. (2006), and Yang et al. (2008). However, when the simulated surface roughness is increased (when $C_p > 0.45$), $g$ values gradually reduce until the ray-tracing results of solid rosettes match the observed $g$ values. Interestingly,

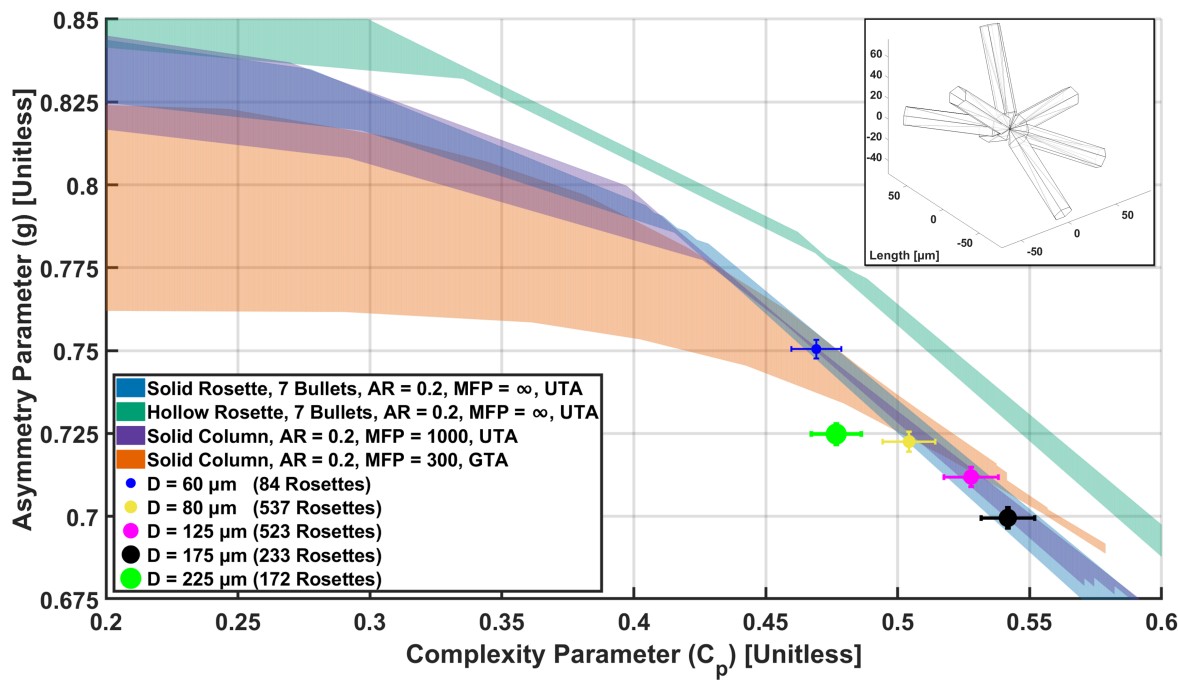

**Figure 10.** Bullet rosette asymmetry parameters ($g$) and their corresponding complexity parameters ($C_\mathrm{p}$) separated by area equivalent diameter ($D$) as measured by the Particle Habit Imaging and Polar Scattering (PHIPS) probe. The values in parentheses indicate the number of rosettes per size bin. Shaded areas show the range from minimum to maximum theoretical results of ray-tracing simulations for solid rosettes, hollow rosettes, and columns. An example modeled hollow rosette can be seen in the upper right, which corresponds to a $D$ of 125 μm.

while increasing the complexity of the simulated hollow rosettes reduces their $g$ values, it does not do so enough to fit the measurements within the bounds of uncertainty.

As can be seen in Table 1, representing the observed in situ data requires a significant amount of simulated roughness ($\delta$ or $\sigma > 0.6$). Previous studies analyzing data measured by a space-borne passive radiometer have indicated $\sigma$ values between 0.15 and 0.7 (Cole et al., 2014; van Diedenhoven, 2021; Järvinen et al., 2023). Baran and Labonnote (2006) concluded that when comparing simulated six-bullet rosette scattering phase functions to those generated from measurements taken by the space-borne passive radiometer POLDER-2, a $\delta$ value of 0.4 was necessary for the simulations to replicate the measurements when considering both total reflectance and polarisation. On the other hand, van Diedenhoven (2021) found roughness parameters ($\sigma$) above 0.6 for cirrus clouds above ocean and land using simulated columns with varying aspect ratios and roughness parameters.

Iaquinta et al. (1995) suggested that bullet rosettes can be represented using simpler models of columns. Such an ability can be desirable as simulations for columns can be completed in shorter times, utilizing less computational resources. Figure 10 also shows simulated $g$ and $C_\mathrm{p}$ values for solid columns with varying surface roughness using both the UTA and GTA methods. The UTA results fit the measured bullet rosette $g$ and $C_\mathrm{p}$ values quite well, nearly perfectly overlapping those of the simulated

**Table 1.** The distortion parameter ($\delta$) and complexity parameter ($\sigma$) resulting in the closest match to the ray-tracing simulated asymmetry parameter ($g$) for solid bullet rosettes, hollow bullet rosettes, and solid columns with the corresponding in situ measured $g$ by area equivalent diameter (D). UTA Solid and hollow rosettes have a have a mean free path (MFP) of infinity. UTA and GTA solid columns have a MFP of 1000 μm and 300 μm. UTA have ($\delta$) ranging from 0.0 to 0.9 increasing by 0.3. GTA have a $\sigma$ range of 0.1 to 0.9, increasing by 0.1.

| Bin Mean Diameter | In Situ Rosettes | UTA Solid Rosettes | UTA Hollow Rosettes | UTA Solid Columns | GTA Solid Columns |
|---|---|---|---|---|---|
| D [μm] | $g$ | $\delta$ | $\delta$ | $\delta$ | $\sigma$ |
| 60 | 0.751 | 0.70 | 0.68 | 0.72 | 0.75 |
| 80 | 0.723 | 0.77 | 0.73 | 0.79 | 0.86 |
| 125 | 0.712 | 0.79 | 0.74 | 0.80 | 0.86 |
| 175 | 0.700 | 0.81 | 0.76 | 0.83 | 0.86 |
| 225 | 0.725 | 0.74 | 0.70 | 0.74 | 0.66 |

solid rosettes. Although the GTA method has a reputation for more sophisticatedly simulating ice crystal complexity, the UTA method produces values that are significantly more representative of the PHIPS measurements. In order for the GTA method to fit within the bounds of uncertainty for even half of the in situ data points, an MFP of 300 is required. Sensitivity tests (not shown here) indicating that higher MFP values generate further unrealistic trends. Although our results confirm the findings of Iaquinta et al. (1995) that the optical behavior ($g$ values) of bullet rosettes can be represented using columns if sufficient surface roughness is included, caution should be applied when using columns as an optical model in climate model applications. Their use may be unsuitable due to inconsistent microphysical coupling - specifically, the inability of the column model to represent mass-dimensional relationships in the microphysics scheme (Baran and Francis, 2004; Ren et al., 2021).

Iaquinta et al. (1995) argued that in order to use columns as an optical proxy for bullet rosettes, their aspect ratio would need to match that of the bullet rosettes. We investigated the sensitivity of $g$ to column aspect ratio and Figure 11 shows the same PHIPS measured results as in Fig. 10, but in relation to theoretical results of ray-tracing simulations for solid columns using the UTA method with $AR$ of 0.1, 0.2, 0.3, and 0.5. It can be seen that with little to no simulated roughness ($C_\text{p} < 0.45$), the difference between simulation results for different aspect ratios is significant. However, with increasing levels of simulated roughness the spread between results narrows. With $C_\text{p} > 0.45$, the largest differences occur in the range of $g = 0.75$ to $g = 0.77$, where there is a maximum percent difference of only 2.2 % between an $AR$ of 0.1 and 0.5. For each of the measured rosette size bins the calculations are well within uncertainty, indicating that with sufficient complexity the relationship between $AR_B$ and the $AR$ of the simulated column is not a critical factor.

Our observations highlight that the largest effect on the $g$ values of bullet rosettes is caused by small scale morphological complexity that is likely comprised of nano-scale surface roughness, stepped patterns within hollow cavities (stepped hollowness), internal inclusions, and/or other deformations rather than micro-scale properties such as $D_{max}$, number of bullets, bullet geometry and $AR_B$. Some evidence of crystal complexity can be clearly seen in Figs. 1 and 3 as a black shading in the stereomicroscopic images; the complexity causes an increase in light diffusion as there is a decease in the amount of light which can

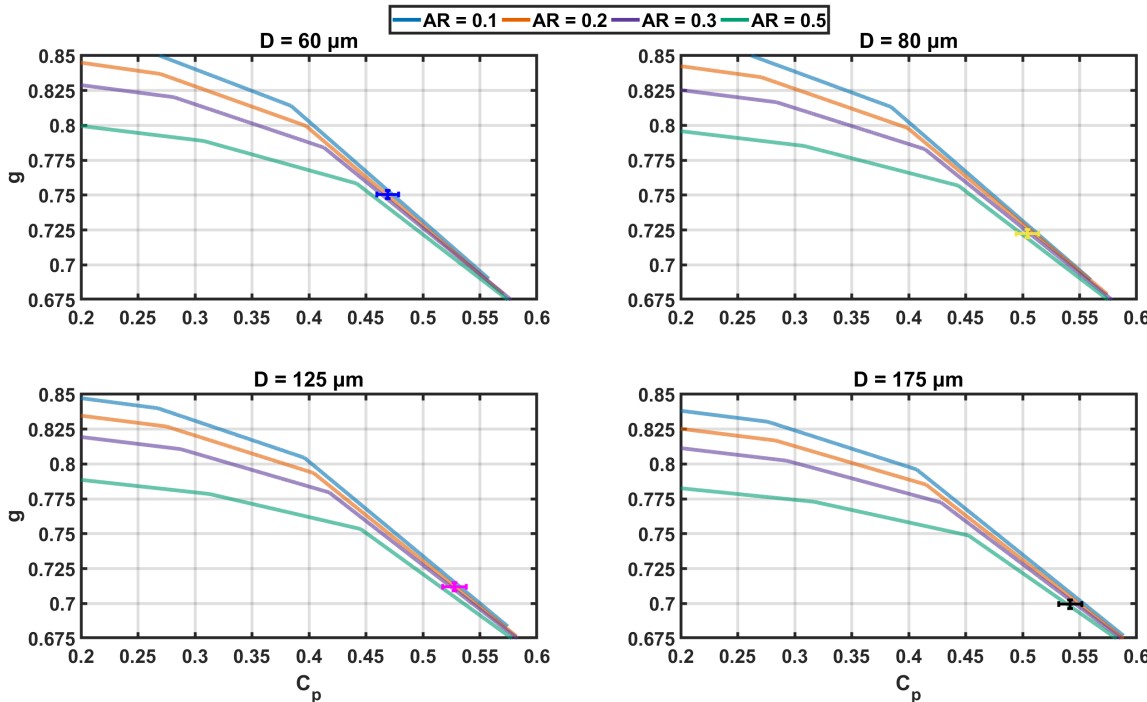

**Figure 11.** The theoretical results of ray-tracing simulations for solid columns with varying aspect ratios ($AR$) using the uniform tilted angle distribution method (UTA) for area equivalent diameters ($D$) ranging from 60 μm to 175 μm compared to the Particle Habit Imaging and Polar Scattering (PHIPS) probe bullet rosette measurements of equal size.

travel undisturbed. These results showcase the necessity of a (single particle) polar nephelometer such as the PHIPS to bridge the gap between numerical simulations and direct measurements of atmospheric crystals.

## 4   Summary and Conclusions

Bullet rosettes commonly occur in high altitude cirrus clouds, and thus play an important role in the radiative budget of the Earth. To properly model and account for the extent of the radiative effect from bullet rosettes, one must understand their single-scattering properties. In this study, we have examined both the microphysical and single scattering properties and their link of real atmospheric bullet rosettes measured by the PHIPS during the CIRRUS-HL airborne campaign. Both the microphysical and single scattering properties were measured on the same individual crystals. We have shown that a cirrus cloud during CIRRUS-HL of temperatures at or below -50 °C is comprised of bullet rosettes on average between 10 and 20 % . The maximum dimensions ranged between 40 μm and 1000 μm with a range of medians from 200 μm to 420 μm μm. The largest maximum dimensions were observed at -35 °C and warmer, the median number of bullets between seven and eight, and a median $H_{FACTOR}$ of approximately 0.75 at temperatures above -35 °C, and which primarily increased with decreasing

temperature to values as high as 1.0. Updated parameterizations for relating rosette maximum dimension to rosette mass, effective density and terminal velocity, as well as the rosette projected area to mass are given, highlighting the importance of accounting for both the number of bullets per rosette as well as the degree of bullet hollowness.

Asymmetry parameters for all rosettes was found to be 0.718, with little difference between solid ($g$ = 0.700), hollow ($g$ = 0.721) and inclusion ($g$ = 0.692) rosettes, with each case lower than previously suggested $g$ based on simulations using rosettes with idealized pristine surfaces. For all rosettes grouped by similar area equivalent diameters ($D$), $g$ was found to decrease with size and range from as low as 0.700 to as high as 0.751, which is lower than previous theoretical studies with a mean percent difference of 11.5 %. The primary cause of the difference between theoretical and actual bullet rosette $g$ values was shown to result from surface level and internal complexity (e.g. surface roughness and stepped hollowness) rather than the microphysical properties (e.g. $D_{max}$, number of bullets, bullet geometry and bullet aspect ratio $AR_B$). This complexity renders assumptions in ray-tracing models that bullet rosettes have idealized pristine surfaces unrealistic, causing the discrepancy between the $g$ values found herein and the results of previous theoretical studies. While some newer climate and weather models include parameterizations for surface roughening to account for observed featureless angular scattering functions (see Um and McFarquhar, 2007; Yang et al., 2013; Baran et al., 2014, 2016), this study has shown that a significant degree of surface roughness ($\delta$ or $\sigma > 0.6$) is needed to represent our in situ observations. It should also be noted that while the ray-tracing simulations of hollow bullet rosettes were found to be outside the bounds of uncertainty of the in situ measured rosettes, the ray-tracing simulations of solid columns and bullet rosettes were within. This supports previous claims that the optical effects of bullet rosettes can be represented with a columnar model at a wavelength of 532 nm, though only if it appropriately accounts for crystal complexity with little importance on the column $AR$.

Work is needed to gauge the full effect of discrepancies between theoretical transfer simulations and measured results of bullet rosette single-scattering properties in radiative transfer simulations, though lower $g$ values in the shortwave lead to more reflected shortwave radiation. When considering scaled optical depth as $\tau_* = (1-g)\tau$ (Liou, 2016; Xu et al., 2022a), the mean percent difference of 11.5 % in the herein measured $g$ values compared to those of previous studies equates to an average 53 % increase in $\tau_*$. In other words, failing to account for the lower $g$ values associated with the complex bullet rosettes would lead to an underestimation in cirrus cloud shortwave reflection by as much as 53 %. While radiative transfer simulations are beyond the scope of this paper, Järvinen et al. (2018) has shown that increased cloud ice crystal complexity (ice crystal $g$ = 0.75) has a significant global climate cooling effect. Therefore, a reduction in $g$ from cloud in situ bullet rosettes caused by crystal complexity, compared to higher $g$ values of previous theoretical studies that assumed idealized pristine surfaces, must be accounted for going forward with climatological and radiative modeling to achieve accurate results.

*Data availability.* Bullet rosette data from the PHIPS probe is available through Zenodo (https://doi.org/10.5281/zenodo.15343726). All processed PHIPS data from CIRRUS-HL, and the atmospheric state data, can be accessed from the HALO database (https://halo-db.pa.op.dlr.de/mission/125).

## Appendix A: PHIPS Data Correction and Uncertainty

One significant factor contributing to uncertainties in retrieving $g$ and $C_{\mathrm{p}}$ is the non-uniform response of the scattering channels to light scattering signals of the same intensity. This non-uniform response can be attributed to potential differences in the coupling of the scattered light to the optical fibers, variations in signal transmission within the optical fibers, dissimilarities in how light is transmitted from fibers to the multi-anode photomultiplier tube (MAPMT), and by the channel-to-channel variation of the anode sensitivity within the MAPMT. This non-uniform response of the scattering channels can be better understood and collectively characterized by measuring particles with a known phase function. For example, using spherical particles with a known size and refractive index.

PHIPS was calibrated during CIRRUS-HL with 81 polystyrene microspheres (Thermofisher, 2024) having diameters of 20 μm and a refractive index of 1.605 + 0.0003i at a wavelength of 532 nm (Jones et al., 2013). Using the size and refractive index, a theoretical angular scattering function was calculated using the Lorenz-Mie theory (Mie, 1908) with an angular resolution of 0.09°. The angular scattering function was then integrated over the solid angles of the PHIPS scattering channels to calculate the expected scattered power per detection angle in nW. Once integrated, a recursive function determines a scaling factor for each microsphere to convert the PHIPS measured counts to nW by minimizing the square residuals per scattering angle between the PHIPS measurements and theoretical calculation. With the conversion factor applied to the PHIPS results, the ratio of the theoretical function to the PHIPS measured function was calculated by channel to quantify the non-uniformity of the PHIPS response and generate channel specific correction factors. Uncertainty in the MAPMT measurements combined with imperfections in the sphericity and size of the microspheres results in a variability of correction factors, and thus the median and standard deviation is taken for each channel.

Figure A1 shows the measured angular scattering function of all 81 polystyrene microspheres and the theoretical function. The blue dashed line is the initial, uncorrected angular scattering function of a randomly selected microsphere. The red dashed line is the theoretical Lorenz-Mie function calculated using the sphere's size and refractive index. The green line is the resulting PHIPS angular scattering function after the multiplicative factors were applied to the measurements of each of the scattering channels. The bottom of Fig. A1 shows the resulting residuals. While the 18 - 162° scattering intensities as measured by the PHIPS match that of the theoretical Lorenz-Mie function for the 20 μm spheres quite well, the Lorenz-Mie function is approximately 1.8 times as intense at the 170° angle. Sensitivity tests on how minor changes of the refractive index affect the resulting Lorenz-Mie calculation (not shown) indicate that the backward angles are the most sensitive to refractive index variations. Variations in the index of refraction combined with imperfections in the microsphere sphericities and surfaces, as well as geometrical scattering effects due to the Gaussian profile of the laser beam, may be the cause of the discrepancy. For further discussions on PHIPS scattering measurements as shown with polystyrene microspheres, piezo generated droplets, and atmospheric ice crystals, see Schnaiter et al. (2018).

Since the discrepancy between PHIPS measurements and the theoretical calculation occurs only in the backward direction, and $g$ is most affected by scattering in the forward direction, correction factors are only applied to angles 18 - 66° where there is no uncertainty caused by deviations in the sphericity. However, even without a correction applied to data, the standard deviation

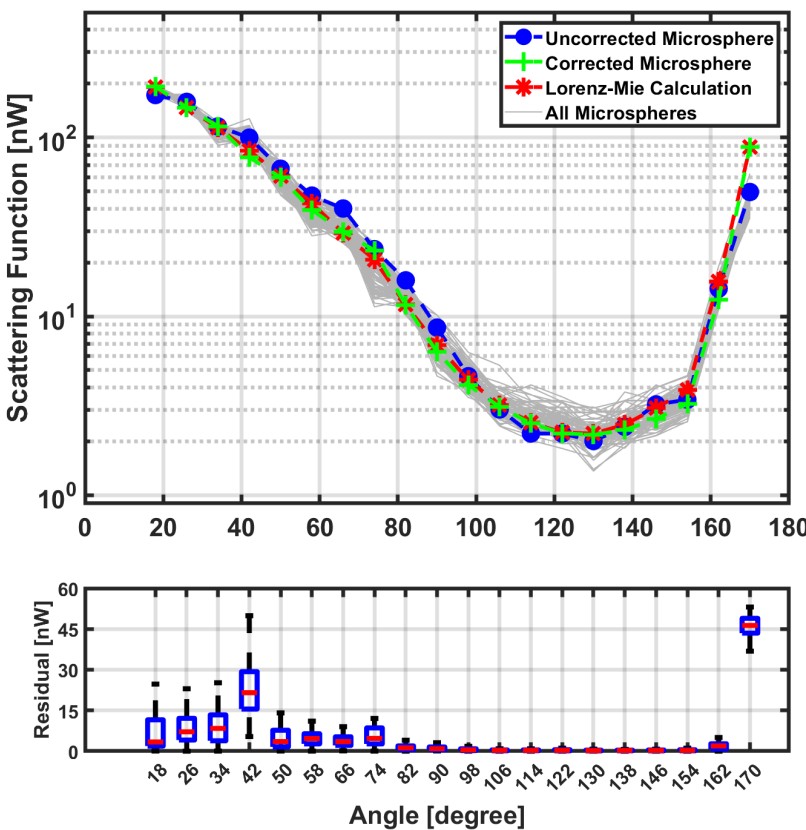

**Figure A1.** An example of one of the 81 microspheres used for the calibration measurements taken by the Particle Habit Imaging and Polar Scattering (PHIPS) probe during the Cirrus in High Latitudes (CIRRUS-HL) airborne campaign (top). Angles measured by the PHIPS (18 - 170°) are corrected using a fit to the theoretical Lorenz-Mie curve. The calculated residuals between the theoretical Lorenz-Mie calculation and the corrected PHIPS channels for each of the 81 microspheres are included (bottom). Only correction factors for angles 18 - 66° are applied to the CIRRUS-HL data.

in the calculated channel correction factors can still be used as an estimate in the channel uncertainty and thus the uncertainty

in the $g$ and $C_{\mathrm{p}}$ calculations. While not shown, a sensitivity test in which correction factors were applied to channels 18 - 170° was conducted, and the effect on $g$ was found to be only 0.07 %, making the occlusion of any correction to angles 74 - 170° negligible. The correction factors and their standard deviations for each channel as applied to the CIRRUS-HL data can be found in Table A1.

**Table A1.** The Particle Habit Imaging and Polar Scattering (PHIPS) scattering data correction factors and their standard deviations as applied to the Cirrus in High Latitudes (CIRRUS-HL) airborne campaign data.

| Scattering Angle | Correction Factor | Standard Deviation |
|---|---|---|
| 18 | 1.0326 | 0.0472 |
| 26 | 0.9452 | 0.0391 |
| 34 | 1.0859 | 0.0723 |
| 42 | 0.7767 | 0.0671 |
| 50 | 1.0348 | 0.1043 |
| 58 | 1.0723 | 0.1352 |
| 66 | 0.9055 | 0.0950 |
| 74 | 1.0 | 0.2875 |
| 82 | 1.0 | 0.1043 |
| 90 | 1.0 | 0.1584 |
| 98 | 1.0 | 0.1204 |
| 106 | 1.0 | 0.1449 |
| 114 | 1.0 | 0.1516 |
| 122 | 1.0 | 0.1179 |
| 130 | 1.0 | 0.1756 |
| 138 | 1.0 | 0.1503 |
| 146 | 1.0 | 0.1303 |
| 154 | 1.0 | 0.1396 |
| 162 | 1.0 | 0.1440 |
| 170 | 1.0 | 0.3183 |

## Appendix B: Bullet Volume Equations

Calculating bullet rosette mass requires the mean bullet volume $(\overline{V_B})$ per rosette. By treating the bullets as hexagonal columns with hexacone caps and accounting for the average hollowness when applicable, $(\overline{V_B})$ can be calculated using:

$$\overline{V_B} = 3ab(\overline{L_B} - \overline{L_{HC}}) + \sqrt{3}\frac{a^2\overline{L_{HC}}}{2} - \sqrt{3}\frac{a^2\overline{H_{FACTOR}L_B}}{2}, \tag{B1}$$

where $a$ is the length of a hexagonal edge, $b$ is the length to the center of a hexagon from the center of the hexagonal edge, $\overline{L_B}$ is the mean bullet length of all bullets that were measured for the rosette, $\overline{H_{FACTOR}}$ is the mean length of the bullet hollowness
for all bullets where hollowness was measurable (when applicable), and $\overline{L_{HC}}$ is the mean hexacone height for all measured bullets for that rosette. Using the basic geometry of a hexagon, $a$ is calculated as:

$$a = \frac{\overline{W_B}}{tan(60)}, \tag{B2}$$

and $b$ is calculated as:

$$b = \frac{\overline{W_B}}{2}. \tag{B3}$$

where $\overline{W_B}$ is the mean width of all bullets that were measured for the rosette. As the bullet rosette caps are often difficult to discern in images containing rosettes with high numbers of bullets, $L_{HC}$ is not directly measured. Instead, this study assumed both an $L_{HC}$ that is 5 % and 20 % of $L_B$ to account for a range of reasonable bullet cap lengths.

To account for length and width errors resulting from $L_B$ and $W_B$ being assessed from bullet rosette projections on 2-D images, a simulated rosette projection with known $L_B$ and $W_B$ is generated with ten random orientations. The $L_B$ and $W_B$ is
495 then calculated manually for comparison to the known values. It is found that $L_B$ tends to be underestimated by 15 % and $W_B$ tends to be underestimated by 0.007 %. The $L_B$ and $W_B$ measured from the 2-D projections are adjusted by adding the 15 % and 0.007 % respectively.

## Appendix C: Rosette Terminal Velocity Equations

Calculating bullet rosette terminal velocity ($V_t$) requires the derivation of air density ($\rho_a$) and the drag coefficient $C_D$. $\rho_a$ can
be calculated using the well known equation:

$$\rho_a = \frac{P}{R_{specific}T}. \tag{C1}$$

where $P$ is the atmospheric pressure in Pa, $R_{specific}$ is the specific gas constant for dry air (287.05 J kg$^{-1}$ K$^{-1}$), and $T$ is the temperature in K. Following McCorquodale and Westbrook (2021), $C_D$ can be calculated as:

$$C_D = C_0 A_{ratio}^{-0.6} \left( 1 + \frac{D_0}{R_e^{\frac{1}{2}}} \right)^2, \tag{C2}$$

where $C_0 = 0.406$ and $D_0 = 6.32$ are both fitting constants, $A_{ratio}$ is the projected area of the bullet rosette divided by the area of the bullet rosette calculated using the maximum dimension, and $R_e$ is the Reynolds number:

$$R_e = \frac{D_0^2}{4} \left[ \left( 1 + \frac{4X^{\frac{1}{2}}}{D_0^2 C_0^{\frac{1}{2}}} \right) - 1 \right]^2. \tag{C3}$$

$X$ is the Best number and is calculated as:

$$X = \frac{2mg_c\rho_a D_{max}^2}{A\eta^2}, \tag{C4}$$

where $\eta$ is the dynamic viscosity of air and can be calculated using the Sutherland-law (White, 2005):

$$\eta = \eta_0 \left( \frac{T}{273} \right)^{\frac{3}{2}} \left( \frac{273 + 111}{T + 111} \right). \tag{C5}$$

$\eta_0$ is a constant $1.716e^{-5}$ N s m$^{-2}$.

*Author contributions.* This bullet rosette study was conceptualized by SWW and EJ. SWW conducted data analysis and led the manuscript writing. MS developed the PHIPS, provided insight into measurement interpretation, and conducted data analysis. SWW, EJ, and MS collected the measurements during CIRRUS-HL. GX provided the code and expertise necessary for the ray-tracing analysis. FN supplied image analysis ice crystal habit identification. EJ provided funding and guidance in analyzing cirrus cloud data and asymmetry parameters, conducted data analysis, and provided general insight. All authors have reviewed, commented on, and approved the manuscript.

*Competing interests.* Martin Schnaiter and Emma Järvinen are members of schnaiTEC GmbH, the PHIPS manufacturer. Martin Schnaiter is employed part-time by schnaiTEC GmbH.

*Acknowledgements.* We would like to thank the members of the CIRRUS-HL field operations for their efforts in acquiring the data utilized herein. This work was funded by the Helmholtz Association's Initiative and Networking Fund (grant agreement no. VH-NG-1531), the Helmholtz Association research program "Atmosphere and Climate".

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
