# Peer review of "Light scattering and microphysical properties of atmospheric bullet rosette ice crystals"

_EGUsphere, 2024_

## Referee Comment (RC1)

Review of "Light scattering and microphysical properties of atmospheric bullet rosette ice crystals" by Wagner et al.

**Overall recommendation**

This study presents unique measurements of bullet rosettes found in ice clouds, obtained using Particle Habit Imaging and Polar Scattering (PHIPS) during the CIRRUS-HL airborne mission. Simultaneous measurements of microphysical properties and angular intensity of scattered light from the same ice crystal provide unique and highly valuable information that was not previously available.

This study delivers an important result: bullet rosettes, and possibly other ice crystals, exhibit smaller asymmetry parameters than those derived from theoretical calculations that assume smooth crystal surfaces. The methodology and analysis employed in this study have both advantages and disadvantages. Although the PHIPS measurement itself is clear and straightforward, there is some confusion regarding the analysis of microphysical properties and their linkage to the measured scattered light and the theoretical calculations. More details are listed below.

Given this, I recommend that this manuscript be published in *Atmospheric Chemistry and Physics* after the concerns listed below are addressed. The specific concerns supporting this recommendation are detailed below.

**Major concerns**

1. lines 101 - 107

The distinction between the 4512 bullet rosettes (BRs) and the subset of 1292 BRs used for microphysical analysis is unclear. The top-left panel appears to show the maximum dimensions of BRs for both the 4512 and 1292 datasets (black dots, median values). In the top-right panel, the aspect ratio ($AR_B$) seems to correspond to the 4512 BRs, while the bottom-left panel (number of bullets) and the bottom-right panel ($H_{FACOR}$) appear to represent the 1292 BRs. Please confirm if this interpretation is correct and clearly indicate this in the manuscript, figures, or figure captions.

Additionally, there is a discrepancy between the BRs used for deriving microphysical properties (4512 or 1292) and the BRs selected for single-scattering properties (1549). Despite the capability of PHIPS to measure both microphysical and scattering properties simultaneously, the study does not establish a clear connection between these measurements. Instead, the single-scattering properties are presented as averages, potentially obscuring important details.

Presenting individual single-scattering measurements of BRs with their corresponding theoretical calculations would enhance clarity and robustness. If averaging is necessary, provide a clear justification for this approach and explain how it impacts the results. Given that the scattering phase function of bullet rosettes is less sensitive to orientation compared to columns, presenting individual measurements should be feasible and could enhance the clarity and robustness of the analysis.

2. line 126

It is unclear whether the hollowness factor is derived for all bullets within the same BR. If the hollowness factors are obtained from a subset of bullets (e.g., three bullets out of a six-branch BR), clarify whether the hollowness factor represents an average of these bullets. Additionally, confirm if each BR is assigned a single hollowness factor. Since PHIPS provides stereo images, explain how these images are used to retrieve the hollowness factors and how these factors are applied to the microphysical property calculations. According to Appendix B, it appears that an identical hollowness factor is applied to all bullets within the same BR. Please provide more detailed explanations in the manuscript to clarify this methodology, as the current description is confusing.

3. Since PHIPS captures stereo images (C1 and C2) for the same target, two sets of microphysical parameters (e.g., maximum dimension, aspect ratio, projected area) are obtained. Clarify how these two sets of parameters are handled in Figures 5-8 and 10. Are both sets shown separately, or are they combined into a single representation? If a single set is shown, explain the selection process for determining which set of parameters to use. Including this clarification in the manuscript will help avoid confusion.

4. line 179, 10,000 crystal orientations

The PHIPS measurements provide angular scattering intensities for BRs with fixed orientations. In contrast, the theoretical simulations assume 10,000 crystal orientations. In Figures 9 and 10, BRs with different geometrical configurations (e.g., bullet size, BR size, aspect ratio, number of bullets, and hollowness factor) are grouped together. This raises concerns about the validity of comparing these averaged measurements with the theoretical calculations. Additionally, the authors state that both solid and hollow rosette simulations are based on a single bullet configuration with seven bullets.

Please clarify whether the comparison between the averaged measurements and theoretical simulations is appropriate, given the differences in geometrical configurations. Discuss how these differences might affect the validity of the results and, if possible, provide an analysis of individual BR measurements to support the conclusions.

5. line 271

The term "retrieved scattering functions (without diffraction)" is ambiguous. Clarify whether these are theoretical calculations or actual measurements. Since the scattering functions extend to 0°, they appear to be theoretical calculations. If this is the case, specify the BR configurations used for these calculations. If the scattering functions are derived from measurements, explain the method for retrieving values outside the PHIPS measurement range, particularly in the forward scattering direction.

6. Figures 9-11

In Figure 9, the retrieved scattering functions are stated to exclude diffraction. Clarify whether the calculated asymmetry parameters ($g$) shown in Figures 10 and 11 also exclude diffraction. The minimal differences in $g$ between Figures 9 and 10 are unexpected, as larger differences are typically observed when diffraction is included or excluded. Please provide a clearer explanation of this aspect and ensure that the figures accurately reflect the impact of diffraction on the asymmetry parameters.

**Minor concerns**

1. Were any aggregates of bullet rosettes identified in the dataset? If such aggregates were observed, a brief mention or discussion of their occurrence and characteristics would be informative. If none were found, a statement clarifying this would be helpful.

2. Throughout the manuscript, ensure consistent use of terminology when referring to the scattering phase function, angular light intensity, and directional light intensity. Using a single, consistent term will enhance clarity and reduce confusion for readers.

3. lines 255-256

The manuscript suggests that incorporating the hollowness factor ($H_{Factor}$) into the mass calculation should decrease the mass. However, the results show an opposite trend. Including a brief explanation in the manuscript would address this inconsistency.

4. Eq. B1

Equation B1 calculates the mean volume of bullets within a bullet rosette. This implies that the bullets within the same BR have different volumes, potentially due to variations in geometrical parameters such as $L_B$ (bullet length), $L_{HC}$ (hollowness length), or the hollowness factor. How did the authors handle these variations when calculating the mean bullet volume? Clarifying the approach used to manage these differences in geometrical configuration would improve the transparency and robustness of the analysis?

This manuscript contains several typographical errors, including those listed below. Additional errors may exist, so a thorough review is recommended to identify and correct them all.

**Typographical and Editorial Corrections**

Line 37: "dimenesion" → "dimension"

Line 71: "duing" → "during"

Line 87: "acquistion" → "acquisition"

Line 103: "Aditionally" → "Additionally"

Line 104: "a criteria" → "a criterion"

Line 105: "occured" → "occurred"

Line 110: Rephrase to: "bullet rosettes have bullets with cavities that begin at..."

Line 125: "assesed" → "assessed"

Line 141: "rossette" → "rosette"

Line 191: "temperatues" → "temperatures"

Line 193: "discrepency" → "discrepancy"

Line 264: "discrepency" → "discrepancy"

Figure 7 Caption: "roesttes" → "rosettes"

Line 323: "signficant" → "significant"

Line 341: "occure" → "occur"

Line 343: "sufficent" → "sufficient"

Lines 358-361: Clarify or correct the term "me" (likely a typo).

Line 418: "discrepency" → "discrepancy"

Line 438: "assesed" → "assessed"

Line 447: "j kg$^{-1}$ K$^{-1}$" → "J kg$^{-1}$ K$^{-1}$"

Page 20: Add missing references for (Thermofisher, 2024) and (Jones et al., 2013).

Figure 4: Correct the vertical axis label to "$AR_B$" instead of "$A_R$".

---

## Referee Comment (RC2)

Review of "Light scattering and microphysical properties of atmospheric bullet rosette ice crystals" by Wagner et al.

Summary

The paper provides a detailed analysis of the microphysical and optical properties of bullet rosette ice crystals, a common habit in in-situ generated cirrus. Using data from the CIRRUS-HL airborne campaign, the authors employ the Particle Habit Imaging and Polar Scattering (PHIPS) probe to link microphysical properties (e.g., bullet dimensions, aspect ratios, and hollow structures) with light-scattering characteristics, particularly the asymmetry parameter (g). The g for bullet rosettes was measured to be consistently lower (mean g=0.718) than previous theoretical predictions, which assumed idealized, pristine surfaces. The authors found that surface complexity, such as roughness and stepped hollowness, was the primary factor causing the discrepancy, with minimal impact from microphysical parameters such as maximum dimensions or number of bullets. The findings suggest that theoretical models overestimate g due to oversimplified assumptions about crystal smoothness, leading to inaccuracies in climate simulations of the radiative effects of cirrus. However, theoretical models that assume some representation of surface roughness on the surfaces of bullet rosettes or aggregates of rosettes (which these days would be most models) are more representative of the authors experimental findings.

The manuscript is well-structured, and the presentation of results is thorough, supported by clear figures. The authors have done well in detailing their methods, validating their results, and contextualizing their findings within the existing literature. My comments are minor and overall, this paper is a strong contribution to atmospheric science and should be considered for publication with the following minor revisions.

Specific comments are listed as follows:

1. Abstract. Line 9. '…optical…' -> 'the optical…'
2. Abstract. Line 10 comment. Most parametrizations of ice optics used in climate models these days include surface roughening effects rather than assuming smooth surfaces. This is done to mimic featureless phase functions which are most often observed. You should acknowledge this in the text of your manuscript. For instance, the ice optical parametrizations of Yang, P., Bi, L., Baum, B. A., Liou, K. N., Kattawar, G. W., Mishchenko, M. I., & Cole, B. (2013). Spectrally consistent scattering, absorption, and polarization properties of atmospheric ice crystals at wavelengths from 0.2 to 100 μm. *Journal of the atmospheric sciences*, 70(1), 330-347; Baran, A. J., Hill P., Furtado K., Field P., and Manners J. (2014). A coupled cloud physics-radiation parameterization of the bulk optical properties of cirrus and its impact on the Met Office Unified Model Global Atmosphere 5.0 Configuration. *J. Climate*, 27, 7725-7752 and Baran, A.J., Hill P., Walters D., Hardiman S. C, Furtado K., Field P. R., and Manners J. (2016). The Impact of Two Coupled Cirrus Microphysics–Radiation Parameterizations on the Temperature and Specific Humidity Biases in the Tropical Tropopause Layer in a Climate Model. *J. Climate*, 29, 5299–5316 – all include surface roughening effects on the surfaces of their ice crystal models to mimic observations to improve radiative simulations in climate and weather models.

3. Introduction. Line 13. On the global percentage distributions of cirrus – there are more updated references. For instance, the well-known works of Stubenrauch, see for instance Stubenrauch, C. J., Feofilov, A. G., Protopapadaki, S. E., and Armante, R.: Cloud climatologies from the infrared sounders AIRS and IASI: strengths and applications, Atmos. Chem. Phys., 17, 13625–13644, https://doi.org/10.5194/acp-17-13625-2017, 2017.

4. Introduction. Line 15. Why are you mentioning 'photons' in the context of atmospheric physics? Since you are applying geometric optics later on in the paper, I suggest you use 'rays'.

5. Introduction. Line 15. Citations of Paltridge and Liou are a bit dated. Suggest you augment these with more updated references such as Yang, P., Liou, K. N., Bi, L., Liu, C., Yi, B., & Baum, B. A. (2015). On the radiative properties of ice clouds: Light scattering, remote sensing, and radiation parameterization. *Advances in Atmospheric Sciences*, 32, 32-63 and Baran, A. J. (2012). From the single-scattering properties of ice crystals to climate prediction: A way forward. *Atmos. Res.*, 112, 45-69.

6. Introduction. Line 19. '…at present…' -> '…at the present…'

7. Introduction. Line 31. Readers might not be conversant with the term 'effective density', please explain the term.

8. Introduction. Line 34. Explain how you define 'maximum dimension' in your paper – how do you measure it from your observations?

9. Introduction. Line 35. Did Fridlind et al. (2016) not provide power laws for their mass derivations. If so, why not quote the power laws instead of numbers?

10. Introduction. Line 44. The paper for comparisons with others relies on the works of Iaquinta et al. (1995) and Schmitt et al. (2006). However, the works of Yang and his students have also produced papers on the single-scattering properties of solid and hollow bullet rosettes. Why have the authors not included the latter works? See for instance, Yang et al. (2008) present differing results to those of Schmitt et al. (2006). Yang, P., et al., (2008). Effect of Cavities on the Optical Properties of Bullet Rosettes: Implications for Active and Passive Remote Sensing of Ice Cloud Properties. *J. Appl. Meteor. Climatol.*, **47**, 2311–2330.

11. In the paper, what is your justification for assuming random orientation? Is this because the quoted comparison citations also assume random orientation or is it for experimental reasons such as noise reduction or both?

12. Section 2. Line 66. '…in base..'-> '..in the base..' also

13. Same line. '….required focus…'-> add 'the' after required.

14. Section 2. Line 90. '…depening…'-> '…depending…'

15. Section 2. Line 93. What does a magnification setting of 4 mean?

16. Line 110. On examination of your Fig 1. I am not convinced that the solid bullet rosette shown contains no air cavities – on closer inspection there does appear to be some faint cavity – bottom-left on both solid images. Can you clarify?

17. Line 125 space between '…to $L_B$ ..'

18. Line 136. 'solid ice' preferred as opposed to just 'ice'.

19. Line 140 '…measure…'-> '…measured…'

20. Same line, the projected area A is this assuming random orientation? If so, use <A>.

21. Line 155. Why is Xu et al. (2022) bracketed?

22. Line 165. The numerical simulations applying a large-scale approximation to surface roughness. Is there any work in the literature that compares this large-scale approximation to more accurate representations of surface roughness in calculating optical properties such as the asymmetry parameter?

23. Line 179. Why 10000 orientations? How do you know whether this is a sufficient number of orientations?

24. Line 238. '…calculate the uncertainty bullet length…'->'….calculate the uncertainty in the bullet length….'

25. Line 249. '…..Fridlind et al. (2016) *becomes* slightly higher…'

26. In this section, the derived mass and area power laws please can you state the units of the maximum dimension? Moreover, another mass-D power law that was derived using in-situ observations of cold cirrus and often used has been derived by Cotton et al. (2013), mass=$0.0257D_{max}^2$ (SI units). See, Cotton, R.J., Field, P.R., Ulanowski, Z., Kaye, P.H., Hirst, E., Greenaway, R.S., Crawford, I., Crosier, J. and Dorsey, J. (2013), The effective density of small ice particles obtained from in situ aircraft observations of mid-latitude cirrus. Q.J.R. Meteorol. Soc., 139: 1923-1934.When using the Brown and Francis (1995) relationship did you correct for their definition of mean diameter to maximum dimension following the Hogan et al. (2012) correction factor? See Hogan, R. J., Tian, L., Brown, P. R. A., Westbrook, C. D., Heymsfield, A. J., and Eastment, J. D.: Radar Scattering from Ice Aggregates Using the Horizontally Aligned Oblate Spheroid Approximation, J. Appl. Meteorol. Clim., 51, 655–671, 2012.

27. Line 269. '….indidual..'->'….individual…'

28. Figure 6. Can you please state the percentage of bullet rosettes that comprised of unmeasurable hollowness?

29. Line 325. The work of Baran and Labonnote (2006) is relevant to this paper because they found that by distorting the six-branched bullet rosette using a distortion parameter of 0.4 they were able to replicate the global POLDER measured polarized reflectances fairly well, see Baran, A. J., & Labonnote, L. C. (2006). On the reflection and polarisation properties of ice cloud. Journal of Quantitative Spectroscopy and Radiative Transfer, 100(1-3), 41-54.

30. Line 328. The discussion about column ice crystals being used to represent bullet rosettes. Since this paper uses one single wavelength, we must be careful not to exaggerate the applicability of simple models to other regions of the electromagnetic spectrum. This is shown quite nicely by Baran and Francis (2004) who found that simple hexagonal columns were an inadequate model relative to a complex aggregate model when differing portions of the observed electromagnetic spectrum were used simultaneously to test the models. See Baran, A.J. and Francis, P.N. (2004), On the radiative properties of cirrus cloud at solar and thermal wavelengths: A test of model consistency using high-resolution airborne radiance measurements. Q.J.R. Meteorol. Soc., 130: 763-778.

31. Figure 8 caption. Space between corresponding and $H_{FACTOR}$.

32. Line 340. '…rougness…'-> '…roughness…'

33. Line 355. In this study,….

34. Line 360. What is 'me'? Seems to be a typo.

35. Line 398. '…caluate..'->'…calculate…'

36. Appendix C. Are the authors aware of the refinements of $C_D$ for bullet rosettes presented by McCorquodale and Westbrook (2021)? See McCorquodale MW, Westbrook CD. TRAIL part 2: A comprehensive assessment of ice particle fall speed parametrisations. *QJR Meteorol Soc*. 2021; 147: 605–626.

---

## Author Response (AR1)

**Egusphere-2024-3316 Authors' Responses to Anonymous Referee #1 and #2**

The authors would like to thank anonymous referees #1 and #2 for providing thoughtful feedback and advice on the manuscript contents. These suggestions have helped to improve the manuscript. All comments have been thoroughly reviewed and incorporated where possible. The comments by the referees are presented first, followed by the authors' responses in *italic font* and the respective changes to the manuscript.

**Referee #1**

**Major concerns**

1. lines 101 – 107 Part 1: The distinction between the 4512 bullet rosettes (BRs) and the subset of 1292 BRs used for microphysical analysis is unclear. The top-left panel appears to show the maximum dimensions of BRs for both the 4512 and 1292 datasets (black dots, median values). In the top-right panel, the aspect ratio ($AR_B$) seems to correspond to the 4512 BRs, while the bottom-left panel (number of bullets) and the bottom-right panel (HFACOR) appear to represent the 1292 BRs. Please confirm if this interpretation is correct and clearly indicate this in the manuscript, figures, or figure captions. Additionally, there is a discrepancy between the BRs used for deriving microphysical properties (4512 or 1292) and the BRs selected for single-scattering properties (1549).

*Response: The interpretation of the maximum dimensions plot is correct. However, $AR_B$ and the number of bullet plots corresponds to the 1292 subset, with the $H_{FACTOR}$ plot representing only hollow rosettes within the 1292 subset which showed a fully observable amount of hollowness. The authors agree that this distinction should be made clearer. Lines 217 – 220 have been updated to state: "These variations can also be seen numerically in Fig. 4, which shows the bullet rosette maximum dimension (top left) for all 4512 successfully imaged bullet rosettes with black dots indicating median values for all 1292 rosettes which were accepted for a further microphysical analysis, the bullet $AR_B$ and number of bullets per rosette for the 1292 microphysically analyzed bullet rosettes, and $H_{FACTOR}$ for the bullet rosettes with observable hollowness in the 1292 rosette subset." The Fig. 4 caption has been updated to state "The bullet rosette maximum dimension (top left) for all 4512 successfully imaged bullet rosettes. Black dots indicate median values for all 1292 rosettes which were accepted for a further microphysical analysis. The bullet $AR_B$ (top right) and number of bullets per rosette (bottom left) for the 1292 bullet rosettes accepted for a microphysical analysis. $H_{FACTOR}$ of individual bullets (bottom right) for rosettes with observable hollowness taken from the 1292 rosette subset ($H_{FACTOR} = 1$ is completely hollow). For each plot bullet rosettes are grouped by temperature."*

1. lines 101 – 107 Part 2: Despite the capability of PHIPS to measure both microphysical and scattering properties simultaneously, the study does not establish a clear connection between these measurements. Instead, the single-scattering properties are presented as averages, potentially obscuring important details. Presenting individual single-scattering measurements of BRs with their corresponding theoretical calculations would enhance clarity and robustness. If averaging is necessary, provide a clear justification for this approach and explain how it impacts the results. Given that the scattering phase function of bullet rosettes is less sensitive to orientation compared to columns, presenting individual measurements should be feasible and could enhance the clarity and robustness of the analysis.

*The focus of this work is to present relevant single-scattering properties for radiative transfer studies of a specific ice crystal habit. The scattering properties are averaged to derive the asymmetry parameter for clouds comprised of bullet rosettes. Calculating an asymmetry parameter for individual rosettes in a single orientation has no practical value for radiative transfer through clouds. PHIPS is unique in its ability to measure individual crystals, allowing for the analysis of single-crystal light scattering by comparison with theoretical calculations. Presenting individual measurements of light scattering for a single ice crystal in a single orientation, which is possible with PHIPS data, offers an interesting aspect for testing theoretical models for a smaller subset of ice crystals. However, such more fundamental studies are not the scope of this work and will be performed in future work. We agree with the reviewer that the presentation of individual scattering function measurements would enhance the clarity of the analysis. Therefore, we overplotted the corresponding scattering functions of the eight individual bullet rosette examples shown in Fig. 3 to the averaged scattering functions in Fig. 9.*

2. line 126 Part 1: It is unclear whether the hollowness factor is derived for all bullets within the same BR. If the hollowness factors are obtained from a subset of bullets (e.g., three bullets out of a six-branch BR), clarify whether the hollowness factor represents an average of these bullets. Additionally, confirm if each BR is assigned a single hollowness factor.

*Response: Hollowness factors are obtained from all bullets in a bullet rosette in which the hollowness is observable, then averaged among those bullets to represent the entire rosette that is being analyzed. All rosettes with observable hollowness are represented with a single hollowness factor that is averaged from their individual analyzable bullets. We have now stated this more clearly in lines: 129 – 132: "The length of the hollowness for each applicable bullet is visually determined via pixel location using the same method applied to the bullet length and width, where the beginning of hollowness is the edge of the bullet, and the end of the hollowness is some point within the bullet. The hollowness factor is then calculated using the method of Schmitt et al. (2006)."*

2. line 126 Part 2: Since PHIPS provides stereo images, explain how these images are used to retrieve the hollowness factors and how these factors are applied to the microphysical property calculations. According to Appendix B, it appears that an identical hollowness factor is applied to all bullets within the same BR. Please provide more detailed explanations in the manuscript to clarify this methodology, as the current description is confusing.

*Response: Images are manually examined using software which allows the user to select the pixel locations of both the beginning and end of the hollow regions. The pixel positions are then used to determine the length of the hollowness. This same process is used to determine the length of the entire bullet for calculating the hollowness factors. It is correct that one hollowness factor is applied to all bullets within the same bullet rosette based on the average. We updated lines 476-479 to include "...$H_{FACTOR}$ is the mean length of the bullet hollowness for all bullets where hollowness was measurable (when applicable) ...". Please see also our response to the following reviewer concern #3 regarding the question on how the stereo images are used to retrieve the hollowness factor.*

3. Since PHIPS captures stereo images (C1 and C2) for the same target, two sets of microphysical parameters (e.g., maximum dimension, aspect ratio, projected area) are obtained. Clarify how these

two sets of parameters are handled in Figures 5-8 and 10. Are both sets shown separately, or are they combined into a single representation? If a single set is shown, explain the selection process for determining which set of parameters to use. Including this clarification in the manuscript will help avoid confusion.

*Response: The rosette maximum dimensions, projected areas, area equivalent diameters, and number of bullets are the maximums of the two cameras, as the largest value is always more representative of the true ice crystal parameters. The bullet widths, lengths, aspect ratios and hollowness factors herein are the mean values to act as representations for each rosette as a whole. The authors agree that this explanation should be added to the text, along with scatter plots to compare the results between cameras one and two. Lines 220 – 223 have been updated to "While the maximum dimensions and number of bullets were taken as the largest values between the two PHIPS cameras because the larger value will always be the most representative, $AR_B$ and $H_{FACTOR}$ are the mean values of the bullets per rosette to act as a representative value." Lines 266 - 267 have been updated to "The rosette projected area is taken as the maximum between the two PHIPS viewing angles." Lines 339 - 340 have been updated to "As done with the maximum dimensions previously, the largest D between the two PHIPS cameras for each rosette is applied."*

4. line 179, 10,000 crystal orientations: The PHIPS measurements provide angular scattering intensities for BRs with fixed orientations. In contrast, the theoretical simulations assume 10,000 crystal orientations. In Figures 9 and 10, BRs with different geometrical configurations (e.g., bullet size, BR size, aspect ratio, number of bullets, and hollowness factor) are grouped together. This raises concerns about the validity of comparing these averaged measurements with the theoretical calculations. Additionally, the authors state that both solid and hollow rosette simulations are based on a single bullet configuration with seven bullets. Please clarify whether the comparison between the averaged measurements and theoretical simulations is appropriate, given the differences in geometrical configurations. Discuss how these differences might affect the validity of the results and, if possible, provide an analysis of individual BR measurements to support the conclusions.

*Response: The PHIPS measured scattering properties herein are averaged over a population of bullet rosettes. A minimum of 50 bullet rosettes was chosen in order to assume random orientation. However, the reviewer is correct that the measured populations varied between 84 and 537, and thus are not comparable to the 10,000 crystal orientations used in the numerical simulations. We investigated how reducing the number of orientations would affect our results. Fig. 1 in this reply shows the results of numerical simulations, where the number of crystal orientations were bound by the number of crystals in the observational data set. Each numerical model was run ten times and Fig. 1 shows the variation in g in comparison to the singular runs with 10,000 orientations. This was repeated for a set of distortion parameters (d). The resulting boxplots using a distortion parameter of 0.6 are included below. The result is that the median of the 10 runs with reduced crystals is nearly identical except for the smallest size (lowest number of orientations) of the hollow rosettes. With the variation in g for the 60 μm size bin having the largest min-max range at 0.011, the variation is negligible. Even less variation is found with the GTA modeled columns. The full collection of boxplots for all habits, area equivalent diameters, and distortion parameters will be included in a supplemental material with the final manuscript. Further discussion on this topic can be found on pages 11 – 13 of this response.*

[Figure]

*Figure 1: Boxplots of the asymmetry parameter (g) versus the area equivalent diameter (D) of modeled rosettes and columns using the Uniform Tilted Angle (UTA) method. Green dots indicate the g values from the singular runs with 10,000 orientations. The values below the boxplots indicate the number of orientations applied to the ten model runs.*

5. line 271: The term "retrieved scattering functions (without diffraction)" is ambiguous. Clarify whether these are theoretical calculations or actual measurements. Since the scattering functions extend to 0°, they appear to be theoretical calculations. If this is the case, specify the BR configurations used for these calculations. If the scattering functions are derived from measurements, explain the method for retrieving values outside the PHIPS measurement range, particularly in the forward scattering direction.

*Response: The lines in Fig. 9 show the retrieved PHIPS measured scattering function using the Legendre polynomials approach, which is a scattering-model-free method to derive phase function from the measurements. This method is applied to extrapolate the measurements beyond the 18 ° and 170 ° detection range of the PHIPS, as well as the interpolation between channels. However, the method cannot be used to reconstruct the diffraction peak at the forward scattering range but quantifies the contribution of diffraction to the combined asymmetry parameter g. Lines 293 -294 have been updated to make this clearer: "The retrieved angular scattering functions are the results of the Legendre polynomial fit applied to the PHIPS measured angular scattering functions."*

*We also added a brief description of the retrieval method to Sec. 2.1.2 to make this clearer: "In short, the method is based on the assumption that the asymmetry parameter in geometrical optics regime can be divided into a diffraction part and geometrical optics part (Macke et al., 1996). The diffraction phase function contributes mainly outside the PHIPS measurement range in the forward direction so that the diffraction g can be estimated based on particle size. The geometrical optics g is derived from the PHIPS measurements by fitting the data with Legendre polynomials. When we later report g, the values represent the combined g, incorporating both the diffraction and geometrical optics components."*

6. Figures 9 – 11: In Figure 9, the retrieved scattering functions are stated to exclude diffraction. Clarify whether the calculated asymmetry parameters (*g*) shown in Figures 10 and 11 also exclude

diffraction. The minimal differences in $g$ between Figures 9 and 10 are unexpected, as larger differences are typically observed when diffraction is included or excluded. Please provide a clearer explanation of this aspect and ensure that the figures accurately reflect the impact of diffraction on the asymmetry parameters.

*Response: The calculated asymmetry parameters in Figs. 9 and 10 include diffraction. To clarify this, we added the sentence "When we later report g, the values represent the combined g, incorporating both the diffraction and geometrical optics components." to Sec. 2.1.2.*

**Minor concerns**

1. Were any aggregates of bullet rosettes identified in the dataset? If such aggregates were observed, a brief mention or discussion of their occurrence and characteristics would be informative. If none were found, a statement clarifying this would be helpful.

*Response: Within the CIRRUS-HL data set, there were 665 bullet rosette aggregates which were fully captured by both camera 1 and 2. Aggregates were most frequently observed with large rosettes and seemingly the result of collisions between bullets. An example of a bullet rosette aggregate from CIRRUS-HL has been included below. While aggregates have been excluded for the purposes of this study, they could be the basis for potential future work. This information has been added to the text. Lines 103 – 105 have been updated to include "Within the CIRRUS-HL data set, there were 665 bullet rosette aggregates which were fully captured by both cameras. Aggregates were most frequently observed with large rosettes and seemingly the result of collisions between bullets. While aggregates have been excluded for the purposes of this study, they could be the basis for potential future work."*

**C1**          **C2**

[Figure]

2. Throughout the manuscript, ensure consistent use of terminology when referring to the scattering phase function, angular light intensity, and directional light intensity. Using a single, consistent term will enhance clarity and reduce confusion for readers.

*Response: The authors agree that it is best for clarity to use consistent terminology. As the most appropriate term as it relates to the PHIPS is "angular scattering function" this term is to be used throughout the manuscript. To account for this, references to the measured functions of the PHIPS have been updated to "angular scattering function".*

3. lines 255 – 256: The manuscript suggests that incorporating the hollowness factor (HFactor) into the mass calculation should decrease the mass. However, the results show an opposite trend. Including a brief explanation in the manuscript would address this inconsistency.

*Response: Incorporating the hollowness factor does decrease mass but the observed increase in mass is due to on average a higher number of bullets per rosette found in this study compared to previous studies. As most other studies assume rosettes with six bullets, and we have found that most rosettes encountered consist of at least 8 bullets. The number of bullets seems to be the dominant factor in the mass calculation rather than if there is hollowness present. Lines 272 – 276 are intended to explain this.*

*As an example, below are images of two rosettes, one classified as hollow ($H_{FACTOR}$ = 0.96), and one classified as solid ($H_{FACTOR}$ = 0) which were included in Fig. 6A. While both have a maximum dimension of approximately 218 μm, the hollow and solid rosette have an equal mass of approximately 1.8 * 10^{-3} mg despite the large $H_{FACTOR}$ of the hollow rosette due to the hollow rosette having 8 bullets while the solid rosette has 7.*

**Hollow**          **Solid**

[Figure]

4. Eq. B1: Equation B1 calculates the mean volume of bullets within a bullet rosette. This implies that the bullets within the same BR have different volumes, potentially due to variations in geometrical parameters such as LB (bullet length), LHC (hollowness length), or the hollowness factor. How did the authors handle these variations when calculating the mean bullet volume? Clarifying the approach used to manage these differences in geometrical configuration would improve the transparency and robustness of the analysis?

*Response: The mean bullet volume is calculated by using the mean length and width of all the bullets that were measured per rosette (and $H_{FACTOR}$ when applicable). So, while it is true that there are variations in length, width, and hollowness between the bullets of an individual rosette, by taking the mean a single representative length, width, and hollowness is determined to calculate the mean bullet volume. Lines 476 – 479 have been updated to "...where a is the length of a hexagonal edge, b is the length to the center of a hexagon from the center of the hexagonal edge, $\overline{L_B}$ is the mean bullet length of all bullets that were measured for the rosette, $\overline{H_{FACTOR}}$ is the mean length of the bullet hollowness for all bullets where hollowness was measurable (when applicable), and $\overline{L_{HC}}$ is the mean hexacone height for all measured bullets for that rosette." Line 483 has been updated to "...where $\overline{W_B}$ is the mean width of all bullets that were measured for the rosette."*

**Typographical and Editorial Corrections**

*All typographical and editorial corrections have been made. It should be noted that Jones et al., 2013 is listed under J. Jones, S., ... as the Middle name is included.*

**Referee #2**

**Minor concerns**

1. Abstract. Line 9. '…optical…' -> 'the optical…'

*Response: Corrected.*

2. Abstract. Line 10 comment. Most parametrizations of ice optics used in climate models these days include surface roughening effects rather than assuming smooth surfaces. This is done to mimic featureless phase functions which are most often observed. You should acknowledge this in the text of your manuscript. For instance, the ice optical parametrizations of Yang, P., Bi, L., Baum, B. A., Liou, K. N., Kattawar, G. W., Mishchenko, M. I., & Cole, B. (2013). Spectrally consistent scattering, absorption, and polarization properties of atmospheric ice crystals at wavelengths from 0.2 to 100 μm. Journal of the atmospheric sciences, 70(1), 330- 347; Baran, A. J., Hill P., Furtado K., Field P., and Manners J. (2014). A coupled cloud physics-radiation parameterization of the bulk optical properties of cirrus and its impact on the Met Office Unified Model Global Atmosphere 5.0 Configuration. J. Climate, 27, 7725-7752 and Baran, A.J., Hill P., Walters D., Hardiman S. C, Furtado K., Field P. R., and Manners J. (2016). The Impact of Two Coupled Cirrus Microphysics–Radiation Parameterizations on the Temperature and Specific Humidity Biases in the Tropical Tropopause Layer in a Climate Model. J. Climate, 29, 5299– 5316 – all include surface roughening effects on the surfaces of their ice crystal models to mimic observations to improve radiative simulations in climate and weather models.

*Response: It is true that more recent bulk optical parameterizations do include surface roughening effects to reproduce the observed featureless phase function. We addressed the reviewer's concern by changing the sentence of lines 9 - 11 of the abstract as follows: "Results indicate that much lower g values represent real atmospheric bullet rosette crystals than what is expected by numerical calculations assuming solid or hollow bullets, indicating higher levels of crystal complexity than have been incorporated within previous bullet rosette ray-tracing studies."*

*We further highlight the reviewer concerns in lines 416 - 419 of the revised manuscript. However, it is important to note here that our observations clearly demonstrate that in-situ light scattering measurements are needed to constrain the degree of surface roughness applied in ice particle optical models. Even if surface roughness is included in those models, the roughness parameters used in the study of Yang et al. are between 0 and 0.5, causing the lowest asymmetry parameters to be around 0.78 – 0.8 for the applicable size range to this study, whereas the complexity parameters we derived are between 0.66 and 0.88 resulting in asymmetry parameters between 0.700 and 0.751 (Tab. 1). This highlights the need for in-situ measurements to aid future model development.*

3. Introduction. Line 13. On the global percentage distributions of cirrus – there are more updated references. For instance, the well-known works of Stubenrauch, see for instance Stubenrauch, C.

J., Feofilov, A. G., Protopapadaki, S. E., and Armante, R.: Cloud climatologies from the infrared sounders AIRS and IASI: strengths and applications, Atmos. Chem. Phys., 17, 13625–13644, https://doi.org/10.5194/acp-17- 13625-2017, 2017.

*Response: After reviewing some of Stubenrauch's work, it was found that an updated global average of 50% is more appropriate. Lines 14 - 15 have been updated to "Cirrus clouds have been found to cover the Earth's surface at a global average of up to 50 % (Wylie et al., 1994; Lynch,15 1996; Sassen et al., 2008; Stubenrauch et al., 2013, 2017)."*

4. Introduction. Line 15. Why are you mentioning 'photons' in the context of atmospheric physics? Since you are applying geometric optics later on in the paper, I suggest you use 'rays'.

*Response: Updated photons to "solar radiation" in line 16.*

5. Introduction. Line 15. Citations of Paltridge and Liou are a bit dated. Suggest you augment these with more updated references such as Yang, P., Liou, K. N., Bi, L., Liu, C., Yi, B., & Baum, B. A. (2015). On the radiative properties of ice clouds: Light scattering, remote sensing, and radiation parameterization. Advances in Atmospheric Sciences, 32, 32-63 and Baran, A. J. (2012). From the single-scattering properties of ice crystals to climate prediction: A way forward. Atmos. Res., 112, 45-69.

*Response: Having reviewed the suggested references, line 17 has been updated to "… (Paltridge, 1980; Liou, 1986; Baran, 2012; Yang et al., 2015)."*

6. Introduction. Line 19. '…at present…' -> '…at the present…'

*Response: Corrected.*

7. Introduction. Line 31. Readers might not be conversant with the term 'effective density', please explain the term.

*Response: Effective density is defined as the ice particle mass divided by the volume of an equivalent diameter sphere. Lines 32 – 34 have been updated to "A number of studies have also been conducted to quantify bullet rosette mass and effective density, which is defined as the particle mass divided by the volume of an equivalent diameter sphere (see Heymsfield et al. 2004) …".*

8. Introduction. Line 34. Explain how you define 'maximum dimension' in your paper – how do you measure it from your observations?

*Response: Maximum dimension is defined as the longest possible axis between two darkened pixels within the camera images. Lines 136 – 138 have been updated to "The maximum dimension ($D_{max}$, defined as the longest possible axis between two darkened pixels) of each bullet rosette is included in the microphysical analysis using IDL image analyzing software that is applied during the primary data processing, and has no corrections applied (Schön et al., 2011)."*

9. Introduction. Line 35. Did Fridlind et al. (2016) not provide power laws for their mass derivations. If so, why not quote the power laws instead of numbers?

*Response: Unfortunately, they did not provide power laws, so values were listed.*

10. Introduction. Line 44. The paper for comparisons with others relies on the works of Iaquinta et al. (1995) and Schmitt et al. (2006). However, the works of Yang and his students have also produced papers on the single-scattering properties of solid and hollow bullet rosettes. Why have the authors not included the latter works? See for instance, Yang et al. (2008) present differing results to those of Schmitt et al. (2006). Yang, P., et al., (2008). Effect of Cavities on the Optical Properties of Bullet Rosettes: Implications for Active and Passive Remote Sensing of Ice Cloud Properties. J. Appl. Meteor. Climatol., 47, 2311–2330.

*Response: The authors would like to thank the review for suggesting Yang et al. (2008) as its exclusion was an oversight. There is a point that is striking in relation to the present study. Yang et al. (2008) has smoother scattering functions than Schmitt et al. (2006), which matches our results. This point has been added to the text. Lines 42 – 43 have been updated to "...such as Iaquinta et al. (1995), Schmitt et al. (2006), and Yang et al. (2008)." Lines 51 – 53 have been updated to "While Yang et al. (2008) reported similar phase functions for solid rosettes as compared to Iaquinta et al. (1995), smoother phase functions for hollow rosettes were seen compared to Schmitt et al. (2006)." Lines 317 – 326 have been updated to "In the sideward and backward direction, the results of Iaquinta et al. (1995), Schmitt et al. (2006), and Yang et al. (2008) are generally similar. However, Yang et al. (2008) found a steeper decrease and a lower flattening of the scattering phase function toward backscattering directions of hollow bullet rosettes with strong hollowness compared to Schmitt et al. (2006). This behavior aligns more closely with the PHIPS-derived functions plotted in Fig. 9. Theoretical results show a general flattening of the angular scattering functions of both solid and hollow rosettes in the sideward directions. In all categories of the PHIPS measured bullet rosettes there is a continued decrease in the angular scattering function until approximately 130° when the trend reverses and a local maximum is reached for all rosettes, hollow rosettes, and inclusion rosettes at 145°. For solid rosettes these features are slightly shifted to 135° and at 155°, respectively. While this local maximum, sometimes referred to as an "ice bow", is readily apparent in Iaquinta et al. (1995) and Yang et al. (2008), it is less apparent in the results of Schmitt et al. (2006) especially for deeply hollow bullet rosettes."*

*In this context we also added a paragraph in lines 311 to 316 of the revised manuscript highlighting our observation of no indications of halo features in the PHIPS data set: "Even though the 22° and 46° halo scattering directions lie precisely between the PHIPS measurement angle pairs 18°/26° and 42°/50°, the absence of any indication of these features in the PHIPS averaged data strongly suggests that they are truly absent in the angular scattering function of real atmospheric bullet rosette ice crystals. This discrepancy can likely be attributed to the theoretical simulations assuming idealized, pristine surfaces while neglecting the natural complexity of ice crystals, a hypothesis that will be further discussed in the following paragraphs."*

11. In the paper, what is your justification for assuming random orientation? Is this because the quoted comparison citations also assume random orientation or is it for experimental reasons such as noise reduction or both?

*Response: As the preferred orientation of the simulated bullet rosettes are not known, the random orientations ensure a statistically robust calculation. This follows the methods of the referenced*

*material such as Um and McFarquhar (2007), Yang et al. (2008), Fridlind et al. (2015), Schmitt et al. (2006), and Iaquinta et al. (1995). Lines 191 – 194 have been updated to include "Randomized orientations provide statistically robust results given that the preferential orientation is not known, and follows the methods of previous studies (Iaquinta et al., 1995; Schmitt et al., 2006; Yang et al., 2008; Um and McFarquhar, 2011; Fridlind et al., 2016)."*

12. Section 2. Line 66. '…in base..'-> '..in the base..' also

*Response: Corrected.*

13. Same line. '….required focus…'-> add 'the' after required.

*Response: Corrected.*

14. Section 2. Line 90. '…depening…'-> '…depending…'

*Response: Corrected.*

15. Section 2. Line 93. What does a magnification setting of 4 mean?

*Response: A "magnification setting of 4" was to indicate the physical setting of the camera magnifications of the PHIPS during CIRRUS-HL. However, this information is unnecessary and potentially confusing, so it has been removed. Lines 96 – 97 have been updated to remove mention of the physical camera setting.*

16. Line 110. On examination of your Fig 1. I am not convinced that the solid bullet rosette shown contains no air cavities – on closer inspection there does appear to be some faint cavity – bottom-left on both solid images. Can you clarify?

*Response: The authors are unable to identify any cavities within the presented solid rosette image. The darkened areas the reviewer refers to rather reflect surface roughness in our opinion. Additionally, there is a small area within the image where the arms are oriented towards the viewer and are slightly out of focus which may seem like they are extending into the underlying bullets.*

17. Line 125 space between '…to LB ..'

*Response: Corrected.*

18. Line 136. 'solid ice' preferred as opposed to just 'ice'.

*Response: Corrected.*

19. Line 140 '…measure…'-> '…measured…'

*Response: Corrected.*

20. Same line, the projected area A is this assuming random orientation? If so, use <A>.

*Response: The projected area is being directly measured for the bullet rosettes from the PHIPS imagery, so no assumption on random orientation is being made.*

21. Line 155. Why is Xu et al. (2022) bracketed?

*Response: This is an error and has been corrected.*

22. Line 165. The numerical simulations applying a large-scale approximation to surface roughness. Is there any work in the literature that compares this large-scale approximation to more accurate representations of surface roughness in calculating optical properties such as the asymmetry parameter?

*Response: Yes, Liu et al. (2014) included below performs a comparison of numerical simulations which show that accurate surface roughening and using irregular facial geometries have similar effects on the scattering of hexagonal columns and can be considered "optically" equivalent. This has been added to the manuscript for future clarity. Lines 182 – 183 have been updated to "Approximations such as the UTA and GTA methods have previously been shown to closely follow more detailed physical representations of surface roughness and are therefore suitable for use herein (Liu et al., 2014)."*

*Liu, C., Panetta, R. L., and Yang, P.: The effective equivalence of geometric irregularity and surface roughness in determining particle single-scattering properties, Opt. Express, OE, 22, 23620–23627, https://doi.org/10.1364/OE.22.023620, 2014.*

23. Line 179. Why 10000 orientations? How do you know whether this is a sufficient number of orientations?

*Response: The PHIPS measured scattering properties herein are averaged over a population of bullet rosettes. A minimum of 50 bullet rosettes was chosen in order to assume random orientation. However, the reviewer is correct that the measured populations varied between 84 and 537, and thus are not comparable to the 10,000 crystal orientations used in the numerical simulations. We investigated how reducing the number of orientations would affect our results. Fig. 1 in this reply shows the results of numerical simulations, where the number of crystal orientations were bound by the number of crystals in the observational data set. Each numerical model was run ten times and Fig. 1 shows the variation in g in comparison to the singular runs with 10,000 orientations. This was repeated for a set of distortion parameters (d). The resulting boxplots using a distortion parameter of 0.6 are included below. The result is that the median of the 10 runs with reduced crystals is nearly identical except for the smallest size (lowest number of orientations) of the hollow rosettes. Shown in Fig. 2 below, the variation in g for the 60 μm size bin has the largest min-max range at 0.011. Therefore, the variation is negligible. Even less variation is found with the GTA modeled columns. The full collection of boxplots for all habits, area equivalent diameters, and distortion parameters will be included in a supplemental material with the final manuscript. Furthermore, Fig. 3 below shows the scattering phase functions as calculated using the UTA method of the modeled solid rosette and solid column which were included in this study for D = 125 μm and d = 0.6 with a range of orientations from 100 to 100,000. While the authors stress that the calculated angular scattering phase function is not compared with the PHIPS data within the study, Fig. 3 below shows that 10,000 orientations result in a smoothing of the angular*

*scattering function which is very similar to 100,000 orientations, with a minor exception at the furthest backward angles. Only the model derived g values are used for comparison with the PHIPS data within the study, which Fig. 1 below clearly shows to be converging to the true value at far fewer orientations than the applied 10,000.*

[Figure]

*Figure 1: Boxplots of the asymmetry parameter (g) versus the area equivalent diameter (D) of modeled rosettes and columns using the Uniform Tilted Angle (UTA) method. Green dots indicate the g values from the singular runs with 10,000 orientations. The values below the boxplots indicate the number of orientations applied to the ten model runs.*

[Figure]

*Figure 2: The difference between the maximum and minimum asymmetry parameters (g) by run versus the area equivalent diameter (D) of modeled rosettes and columns using the Uniform Tilted Angle (UTA) method. The values below the boxplots indicate the number of orientations applied to the ten model runs.*

[Figure]

*Figure 3: The theoretical geometric optics phase functions (excluding diffraction) for the modeled solid rosette and solid column, where D = 125 μm and d = 0.6 with a range of orientations from 100 to 100,000. Calculations were performed using the Uniform Tilted Angle (UTA) method.*

24. Line 238. '…calculate the uncertainty bullet length…'->'….calculate the uncertainty in the bullet length….'

*Response: Corrected.*

25. Line 249. '…..Fridlind et al. (2016) becomes slightly higher…'

*Response: Corrected.*

26. In this section, the derived mass and area power laws please can you state the units of the maximum dimension? Moreover, another mass-D power law that was derived using in-situ observations of cold cirrus and often used has been derived by Cotton et al. (2013), mass=0.0257Dmax2 (SI units). See, Cotton, R.J., Field, P.R., Ulanowski, Z., Kaye, P.H., Hirst, E., Greenaway, R.S., Crawford, I., Crosier, J. and Dorsey, J. (2013), The effective density of small ice particles obtained from in situ aircraft observations of mid-latitude cirrus. Q.J.R. Meteorol. Soc., 139: 1923-1934.When using the Brown and Francis (1995) relationship did you correct for their definition of mean diameter to maximum dimension following the Hogan et al. (2012) correction factor? See Hogan, R. J., Tian, L., Brown, P. R. A., Westbrook, C. D., Heymsfield, A. J., and Eastment, J. D.: Radar Scattering from Ice Aggregates Using the Horizontally Aligned Oblate Spheroid Approximation, J. Appl. Meteorol. Clim., 51, 655–671, 2012.

*Response: The maximum dimension has units of μm, which has now been stated in lines 261 - 262 "...with respect to rosette maximum dimension $D_{max}$ in μm…". Fig. 6 has been updated to include the power law provided by Cotton et al. 2013. Initially a correction was not applied to the Brown and Francis 1995 power law. This has now been included and mentioned in the Fig. 6 caption: "Note that a correction to the Brown and Francis (1995) power law been applied to account for a conversion to maximum dimension according to Hogan et al. (2012)."; however, the result is negligible.*

27. Line 269. '….indidual..'->'….individual…'

*Response: Corrected.*

28. Figure 6. Can you please state the percentage of bullet rosettes that comprised of unmeasurable hollowness?

*Response: Of the 1292 bullet rosettes available for a microphysical analysis, 932 were determined to be hollow. Of those 932, 329 rosettes contained hollow cavities of which the length was able to be fully measured. Therefore, 603 or 65 % had unmeasurable hollowness. Lines 251 – 253 have been updated to "Of the 1292 bullet rosettes chosen for a manual bullet analysis, 932 contained some degree of hollowness. Of those 932, 329 (35 %) have at least one bullet which is not only hollow but has the full extent of the hollowness observable within its associated image."*

29. Line 325. The work of Baran and Labonnote (2006) is relevant to this paper because they found that by distorting the six-branched bullet rosette using a distortion parameter of 0.4 they were able to replicate the global POLDER measured polarized reflectances fairly well, see Baran, A. J., & Labonnote, L. C. (2006). On the reflection and polarisation properties of ice cloud. Journal of Quantitative Spectroscopy and Radiative Transfer, 100(1-3), 41-54.

*Response: The authors thank Reviewer 2 for pointing out this study, and it is interesting that they were able to achieve a match with POLDER-2 with distortion of only 0.4. Their findings have been included in the text. Lines 362 – 366 have been updated to "Previous studies analyzing data measured by a space-borne passive radiometer have indicated σ values between 0.15 and 0.7 (Cole et al., 2014; van Diedenhoven, 2021; Järvinen et al., 2023). Baran and Labonnote (2006) concluded that when comparing simulated six-bullet rosette scattering phase functions to those generated from measurements taken by the space-borne passive radiometer POLDER-2, a δ value of 0.4 was necessary for the simulations to replicate the measurements when considering both total reflectance and polarisation. On the other hand, van Diedenhoven (2021) found roughness parameters (σ) above 0.6 for cirrus clouds above ocean and land using simulated columns with varying aspect ratios and roughness parameters."*

30. Line 328. The discussion about column ice crystals being used to represent bullet rosettes. Since this paper uses one single wavelength, we must be careful not to exaggerate the applicability of simple models to other regions of the electromagnetic spectrum. This is shown quite nicely by Baran and Francis (2004) who found that simple hexagonal columns were an inadequate model relative to a complex aggregate model when differing portions of the observed electromagnetic spectrum were used simultaneously to test the models. See Baran, A.J. and Francis, P.N. (2004), On the radiative properties of cirrus cloud at solar and thermal wavelengths: A test of model consistency using high-resolution airborne radiance measurements. Q.J.R. Meteorol. Soc., 130: 763-778.

*Response: This is an excellent point and the text has been updated to make clear that this applies specifically to the 532 nm wavelength utilized by the PHIPS. Lines 374 – 378 have been updated to "Although our results confirm the findings of Iaquinta et al. (1995) that the optical behavior (g values) of bullet rosettes can be represented using columns if sufficient surface roughness is included, caution should be applied when using columns as an optical model in climate model applications. Their use may be unsuitable due to inconsistent microphysical coupling - specifically, the inability of the column model to represent mass-dimensional relationships in the microphysics scheme (Baran and Francis, 2004; Ren et al., 2021)."*

31. Figure 8 caption. Space between corresponding and HFACTOR.

*Response: Corrected.*

32. Line 340. '…rougness…'-> '…roughness…'

*Response: Corrected.*

33. Line 355. In this study,….

*Response: Corrected.*

34. Line 360. What is 'me'? Seems to be a typo.

*Response: This has been corrected to reflect the median hollowness factor.*

35. Line 398. '…caluate..'->'…calculate…'

*Response: Corrected.*

36. Appendix C. Are the authors aware of the refinements of CD for bullet rosettes presented by McCorquodale and Westbrook (2021)? See McCorquodale MW, Westbrook CD. TRAIL part 2: A comprehensive assessment of ice particle fall speed parametrisations. QJR Meteorol Soc. 2021; 147: 605–626.

*Response: The authors were not aware of the improvements made to $C_D$, but have now adjusted the calculation accordingly. The applied improvements resulted in a decrease of the calculated terminal velocity; however, the terminal velocity is still higher than previous studies as expected with the increase in mass. Accordingly, Fig. 8 has been updated with new calculation, and lines $282 - 283$ have been updated to "Calculated $V_t$ ranges from 60 to 310 cm s−1 for $D_{max}$ between 100 and 900 μm; approximately 50 % higher than the magnitude of values reported by Fridlind et al. (2016)." Equation C2 has been adjusted to match that of McCorquodale and Westbrooke (2021), lines $498 - 499$ have been updated to "where $C_0= 0.406$ and $D_0 = 6.32$ are both fitting constants, $A_{ratio}$ is the projected area of the bullet rosette divided by the area of the bullet rosette calculated using the maximum dimension, ..."*

**Other Changes**

*Lines $74 - 77$ have been updated to "By isolating the angular scattering functions attributed to a specific habit or microphysical feature, it is possible to derive the representative orientation averaged angular scattering function of this specific habit. In essence, this analysis method generates a cloud composed solely of ice crystals with a specific habit or microphysical feature (e.g., bullet rosettes) based on data from real atmospheric ice crystals."*

*Lines $98 - 102$ have been updated to "Of the 5668 total bullet rosettes encountered during CIRRUS-HL, 4512 rosettes are accepted for this study. These bullet rosettes were entirely captured in at least one of the two PHIPS camera focal planes allowing for measurement of the maximum dimension with a high confidence. Of the 4512 bullet rosettes accepted, 1292 are found to have individually identifiable and distinguishable bullets, allowing for further analysis of the bullet related microphysical properties to be performed. These 1292 bullet rosettes will be the primary focus of the microphysical results and discussion."*

---

## Author Response (AR2)

**Egusphere-2024-3316 Authors' Responses to the Editor**

The authors would like to thank the editor for their thoughtful review and suggestions for the manuscript. The suggestions increase help emphasize the importance of the contents and will. The comment that was provided by the editor is included first, followed by the authors' response in *italic font* and the respective changes to the manuscript.

**Editor comment:**

Thank you for your thorough response to the two thorough reviews of your excellent manuscript. Most of the modifications were technical.

I have a couple of things to request, so that the significance of the article can be more widely appreciated. Both in the abstract and conclusions, I would like to see a numerical contrast of the observational results to those more widely assumed. Also, because the asymmetry parameter is not as widely appreciated in its significance relative to other single scattering parameters, I would like to see some statement of the impact of this contrast on the scaled optical depth. It is not so much g that is relevant but rather (1 - g). It appears to me that the results you propose imply possible modeling errors for the shortwave reflection by thin cirrus of order 50%, which is an important point to make, again in the abstract and conclusions.

With these minor adjustments, I will accept the paper.

Best regards,

Tim Garrett

*Response: Adding information on the effect of reduced g values regarding the scaled optical depth ($\tau_*$) helps further emphasize the importance of this work. In calculating $\tau_* = (1-g)\tau$, there is an average 53 % increase when applying the g values determined in this study (ranging from 0.700 – 0.751) as opposed to the g values presented in the discussed previous publications (ranging from 0.750 – 0.876). To incorporate this face, lines 12 – 13 of the abstract have been updated to include " Measured g values herein have a direct impact on modeling the shortwave reflection of cirrus clouds, resulting in an increase in scaled optical depth by an average 53 % in comparison to previously calculated g values." Furthermore, the conclusions have been updated at lines 429 – 432 have been updated to include "When considering scaled optical depth as $\tau* = (1 - g)\tau$ (Liou, 2016; Xu et al., 2022a), the mean percent difference of 11.5 % in the herein measured g values compared to those of previous studies equates to an average 53 % increase in $\tau*$. In other words, failing to account for the lower g values associated with the complex bullet rosettes would lead to an underestimation in cirrus cloud shortwave reflection by as much as 53 %."*